# Glycosylation and stabilization of programmed death ligand-1 suppresses T-cell activity

Chia-Wei Li[1,*], Seung-Oe Lim[1,*], Weiya Xia[1], Heng-Huan Lee[1], Li-Chuan Chan[1,2], Chu-Wei Kuo[3,4], Kay-Hooi Khoo[4], Shih-Shin Chang[1,2], Jong-Ho Cha[1,5], Taewan Kim[1], Jennifer L. Hsu[1,6,7], Yun Wu[8], Jung-Mao Hsu[1], Hirohito Yamaguchi[1], Qingqing Ding[1], Yan Wang[1], Jun Yao[1], Cheng-Chung Lee[3], Hsing-Ju Wu[6], Aysegul A. Sahin[8], James P. Allison[9], Dihua Yu[1,2], Gabriel N. Hortobagyi[10] & Mien-Chie Hung[1,2,6,7]

Extracellular interaction between programmed death ligand-1 (PD-L1) and programmed cell death protein-1 (PD-1) leads to tumour-associated immune escape. Here we show that the immunosuppression activity of PD-L1 is stringently modulated by ubiquitination and N-glycosylation. We show that glycogen synthase kinase 3β (GSK3β) interacts with PD-L1 and induces phosphorylation-dependent proteasome degradation of PD-L1 by β-TrCP. In-depth analysis of PD-L1 N192, N200 and N219 glycosylation suggests that glycosylation antagonizes GSK3β binding. In this regard, only non-glycosylated PD-L1 forms a complex with GSK3β and β-TrCP. We also demonstrate that epidermal growth factor (EGF) stabilizes PD-L1 via GSK3β inactivation in basal-like breast cancer. Inhibition of EGF signalling by gefitinib destabilizes PD-L1, enhances antitumour T-cell immunity and therapeutic efficacy of PD-1 blockade in syngeneic mouse models. Together, our results link ubiquitination and glycosylation pathways to the stringent regulation of PD-L1, which could lead to potential therapeutic strategies to enhance cancer immune therapy efficacy.

[1] Department of Molecular and Cellular Oncology, The University of Texas MD Anderson Cancer Center, Houston, Texas 77030, USA. [2] Graduate School of Biomedical Sciences, The University of Texas Health Science Center at Houston, Houston, Texas 77030, USA. [3] Core Facilities for Protein Structural Analysis, Academia Sinica, Taipei 115, Taiwan. [4] Institute of Biological Chemistry, Academia Sinica, Taipei 115, Taiwan. [5] Tumor Microenvironment Global Core Research Center, College of Pharmacy, Seoul National University, Seoul 151-742, Korea. [6] Center for Molecular Medicine and Graduate Institute of Cancer Biology, China Medical University, Taichung 404, Taiwan. [7] Department of Biotechnology, Asia University, Taichung 413, Taiwan. [8] Department of Pathology, The University of Texas MD Anderson Cancer Center, Houston, Texas 77030, USA. [9] Department of Immunology, The University of Texas MD Anderson Cancer Center, Houston, Texas 77030, USA. [10] Department of Breast Medical Oncology, The University of Texas MD Anderson Cancer Center, Houston, Texas 77030, USA. * These authors contributed equally to this work. Correspondence and requests for materials should be addressed to M.-C.H. (email: mhung@mdanderson.org).

Programmed death ligand-1 (PD-L1), also known as B7 homologue 1 or B7-H1, is a 33 kDa type 1 transmembrane protein that binds to its receptor, programmed cell death protein-1 (PD-1), which leads to the inhibition of T lymphocyte proliferation, cytokine production, and cytolytic activity and suppression of the body's immune response[1,2]. Ligation of PD-L1 and PD-1 receptor complex hijacks the activation state of CD28/MHC with a higher affinity to inhibit T lymphocyte proliferation, cytokine production and cytolytic activity[2]. Monoclonal antibodies blocking this inhibitory pathway-reactivate T-cell activity against cancer cells[3]. Promising clinical outcomes in trials testing antibody blockade of the PD-1/PD-L1 pathway in melanoma[4], lung cancer[5] and kidney cancer[6] have resulted in new treatment options for a broad spectrum of malignant cancers[7].

N-glycosylation plays a critical role in determining protein structure and function. In particular, glycosylation of membrane receptor proteins is important for protein–protein interactions, such as those between ligands and receptors, and has been shown to affect protein activities[8]. After initiation in the endoplasmic reticulum, protein N-glycosylation continues in the Golgi apparatus[9]. This type of post-translational modification is first catalysed by a membrane-associated oligosaccharyl transferase complex that transfers a preformed glycan composed of oligosaccharides to an asparagine (Asn) side-chain acceptor located within the NXT motif (-Asn-X-Ser/Thr-)[10,11] and can be classified into high-mannose, hybrid and complex types based on the saccharide composition and branching features. N-glycosylation affects a number of biological processes, including the modulation of protein–protein interactions[12]. Therapeutically, small-molecule glycan-based drugs have been used as antiviral agents[13,14]. Carbohydrate-based vaccines consisting of glycosylated tumour antigen have also been shown to induce humoral and adaptive response[15]. Most recently, targeting the N-glycan structure of vascular endothelial growth factor (VEGF) receptor was shown to sensitize anti-VEGF refractory tumours to VEGF inhibition[16].

Glycogen synthase kinase 3β (GSK3β) is a serine/threonine protein kinase that was originally identified as a regulator of glycogen metabolism. It was later shown to be a key component of the Wnt signalling pathway, which plays important roles in embryonic development and tumorigenesis[17,18]. GSK3β acts as a multifunctional switch through direct phosphorylation of a wide range of substrates, including eIF2B, cyclin D1, c-Jun, c-myc, NFAT, MCl-1 and Snail[17–20]. GSK3β − mediated phosphorylation often facilitates ubiquitin E3 ligase recognition. For example, GSK3β phosphorylates β-catenin, incorporating β-TrCP for protein degradation. Inhibition of GSK3β allows β-catenin to translocate to the nucleus, form complexes with T-cell factor/lymphoid enhancer factor and thereby activates target gene expression[21].

## Results

**PD-L1 is glycosylated in cancer cells**. While examining PD-L1 protein expression in human tumour tissues and cancer cell lines, we noticed that the majority of PD-L1 was detected at ∼45 kDa (black circle), but a smaller fraction at 33 kDa (arrowhead) also appeared (Fig. 1a,b and Supplementary Fig. 1a,b). Knocking down PD-L1 by lentiviral short-hairpin RNA (shRNA) targeting either the coding sequence (shPD-L1#1) or the 3'-untranslated repeat (shPD-L1#5) downregulated the expression of both the 33- and 45-kDa forms of PD-L1 (Fig. 1c and Supplementary Fig. 1c). Reconstitution of PD-L1 restored the expression of both forms in the shPD-L1#5 clone (Fig. 1c and Supplementary Fig. 1d, vector design). Consistently, CRISPR/Cas9-mediated PD-L1 knockout depleted the 45-kDa form (Fig. 1c, right panel).

Together, these results suggest that both bands represent PD-L1 and that the higher band may be attributed to post-translational modifications.

Glycosylation of proteins often results in a heterogeneous pattern on western blots, such as that observed for PD-L1 (∼45 kDa). Thus, to determine whether this pattern corresponds to PD-L1 glycosylation, we treated MDA-MB-231 and BT549 cells with recombinant glycosidase (peptide-N-glycosidase F; PNGase F) to remove the entire N-glycan structure and then subjected the cell lysates to western blot analysis. A significant portion of the 45-kDa PD-L1 was reduced to 33 kDa upon PNGase F treatment (Fig. 1d). Consistently, positive staining of the glycan structure was observed in purified His-tagged PD-L1 but not in the presence of PNGase F (Supplementary Fig. 1e). These results demonstrate that PD-L1 with the higher molecular weight is indeed the glycosylated form.

Next, to validate PD-L1 glycosylation, we generated plasmids with various tags and evaluated their expression by western blotting. Similar to endogenous PD-L1, Flag-, haemagglutinin (HA)- and green fluorescent protein (GFP)-tagged PD-L1 had an ∼15-kDa molecular weight shift from their actual size when treated with PNGase F (Fig. 1e). Interestingly, the addition of recombinant glycosidase, endoglycosidase H (Endo H), which cleaves high-mannose and some hybrid oligosaccharides, only partially reduced PD-L1 glycosylation, suggesting that the complex type of N-linked glycan structures exists predominantly on PD-L1 (ref. 22). Glycosylation of endogenous and exogenous PD-L1 was completely inhibited when cells were treated with the N-linked glycosylation inhibitor tunicamycin (TM; Supplementary Fig. 2a–c) in a dose-dependent manner but not when they were treated with O-glycosidase (Supplementary Fig. 2d). Together, these results indicate that PD-L1 is extensively N-glycosylated.

Western blot analysis using the PD-L1 extracellluar domain (ECD)- or intracellular domain-specific antibodies indicated that PD-L1 glycosylation occurred on its ECD (Supplementary Fig. 2b). To pinpoint the glycosylation sites, we searched for evolutionarily conserved NXT motifs[9] in the PD-L1 amino-acid sequences from different species. Consistent with the earlier prediction[23], four NXT motifs were identified (Supplementary Fig. 3 and Fig. 1f). To identify the glycan structure, we analysed the tryptic peptides of purified human PD-L1 by nanoscale liquid chromatography coupled to tandem mass spectrometry (LC-MS/MS). Glycopeptides carrying N-glycans, including the complex type, were identified for each of the 4 N-glycosylation sites (Fig. 1g and Supplementary Fig. 4a–f), supporting the apparent resistance to Endo H treatment (Fig. 1e). Substitution of each of the four asparagines (N) to glutamine (Q)—N35Q, N192Q, N200Q or N219Q—led to a certain degree of reduction in glycosylation compared with the wild-type (WT) PD-L1 (Fig. 1h, lanes 2–5). No detectable differences in glycosylation were observed for the three non-NXT NQ PD-L1 mutants (Fig. 1h, lanes 11–13). Interestingly, PD-L1 glycosylation was completely ablated in the PD-L1 4NQ as indicated by the absence of signals corresponding to the non-glycosylated form upon TM treatment (Fig. 1h, lanes 10 and 14). Together, the results indicate that PD-L1 is exclusively N-glycosylated at N35, N192, N200 and N219.

**Glycosylation of PD-L1 stabilizes PD-L1**. Because the levels of glycosylated PD-L1 were significantly higher than the levels of its non-glycosylated form (Fig. 1a,b and Supplementary Fig. 2a–c), we next sought to determine whether glycosylation affects PD-L1 stability. In the presence of protein synthesis inhibitor cycloheximide, the turnover rate for non-glycosylated PD-L1 was faster than glycosylated PD-L1 (Fig. 2a). Similar results were

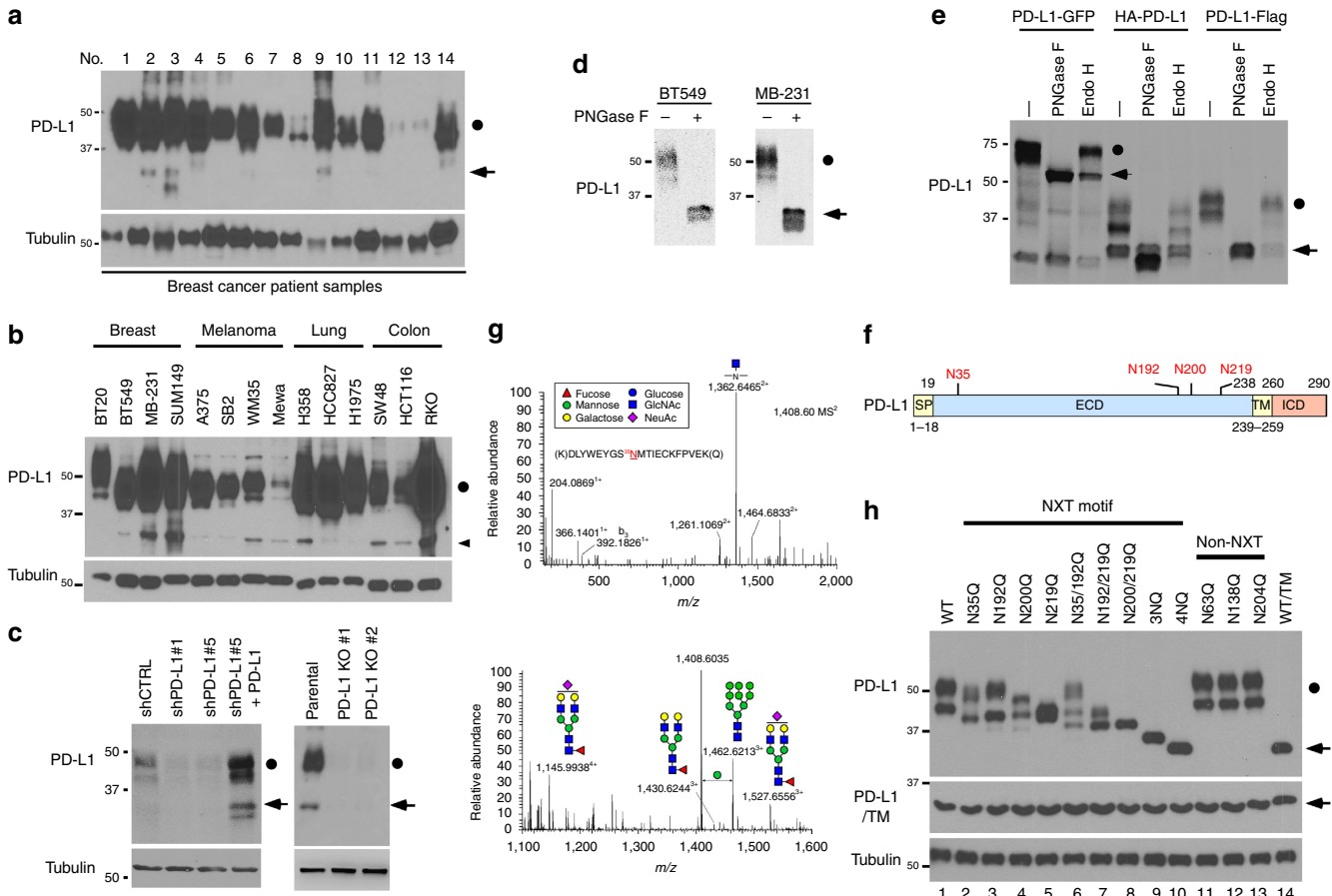

**Figure 1 | PD-L1 is glycosylated in cancer cells. (a)** Expression of PD-L1 protein in primary breast cancer patient samples. Western blot analysis of PD-L1 in representative breast cancer patient samples. **(b)** Western blot analysis of PD-L1 in four breast cancer, four melanoma and three lung and three colon cancer cells. **(c)** Western blot analysis of PD-L1 expression in shCTRL, two independent shPD-L1 stable clones and reconstitution of PD-L1 restored expression in the shPD-L1#5 clone of BT549 cells (left). Western blot analysis of PD-L1 expression in CRISPR/Cas9-mediated PD-L1 knockout BT549 cells (right panel). **(d)** Glycosylation pattern of PD-L1 protein in BT549 and MDA-MB-231 cells. Cell lysates were treated with PNGase F and analysed by western blot analysis. Black circle, glycosylated PD-L1; arrowhead, non-glycosylated PD-L1. **(e)** Glycosylation pattern of PD-L1-GFP, HA-PD-L1 and PD-L1-Flag proteins. Cell lysates were treated with PNGase F and Endo H and analysed by western blot analysis. **(f)** Schematic diagram of PD-L1 protein. Full-length PD-L1 was separated into extracellular domain (ECD) and intracellular domain (ICD). SP, signal peptide; TM, transmembrane domain. Four putative NXT motifs in the ECD domain are labelled in red. The numbers indicate amino-acid positions. **(g)** LC-MS/MS-based identification of N-glycopeptides. LC-MS/MS-based identification of N-glycopeptides corresponding to one of the four N-glycosylation sites, N35. The LC-MS profiles (top) are shown as spectra averaged over a period of elution time (as labelled in figures) when a representative subset of glycoforms were detected. For each N-glycosylation site, one representative HCD MS$^2$ spectrum (bottom) is shown to exemplify its identification based on detection of y1 ion (tryptic peptide backbone carrying the GlcNAc attached to the N-glycosylated Asn), along with the b and y ions defining its peptide sequence. The cartoon symbols used for the glycans (see inset) conform to the standard representation recommended by the Consortium for Functional Glycomics. **(h)** Western blot analysis of the protein expression pattern of PD-L1 WT and its NQ mutants. Non-glycosylated form in lane 14 indicates PD-L1 WT with overnight treatment with TM. Black circle, glycosylated PD-L1; arrowhead, non-glycosylated PD-L1.

observed with overexpressed PD-L1 in HEK 293T cells (Fig. 2b). To test the involvement of 26S proteasome machinery, we subsequently treated TM-treated cells with proteasome inhibitor MG132. We found that only non-glycosylated PD-L1 exhibited more ubiquitination in the presence of MG132 (Fig. 2c and Supplementary Fig. 5a), suggesting that non-glycosylated PD-L1 undergoes fast protein degradation. To pinpoint the exact NXT motif on PD-L1 that is responsible for stabilization, we sought to determine the protein stability of PD-L1 variants (Fig. 2d). The turnover rate of the N35Q mutant was similar to that of WT, suggesting that N35 glycosylation may not be involved in PD-L1 protein stability. Interestingly, N192Q, N200Q and N219Q reduced PD-L1 stability to a certain degree. When these three sites were mutated simultaneously (N35/3NQ), the protein half-life became similar to that of 4NQ (∼4 h; Fig. 2e,f). In the

presence of MG132, PD-L1 N35/3NQ and 4NQ exhibited similar levels of ubiquitination (Fig. 2g). Together, these results suggest that the glycosylation of the N192, N200 and N219, but not N35, contributes to PD-L1 protein stability.

**GSK3β binds to and phosphorylates PD-L1.** Next, we asked how N192, N200 and N219 glycosylation might regulate PD-L1 protein stability. We noted that PD-L1 contains a consensus GSK3β phosphorylation motif (SxxxTxxxS, in which S represents serine, x represents any amino acid and T represents threonine) adjacent to N192, N200 and N219 glycosylation sites (Fig. 3a, which were critical for PD-L1 stability (Fig. 2d–g)). Protein sequence alignment demonstrated that the phospho-motif is evolutionarily conserved (Supplementary Fig. 5b). GSK3β-

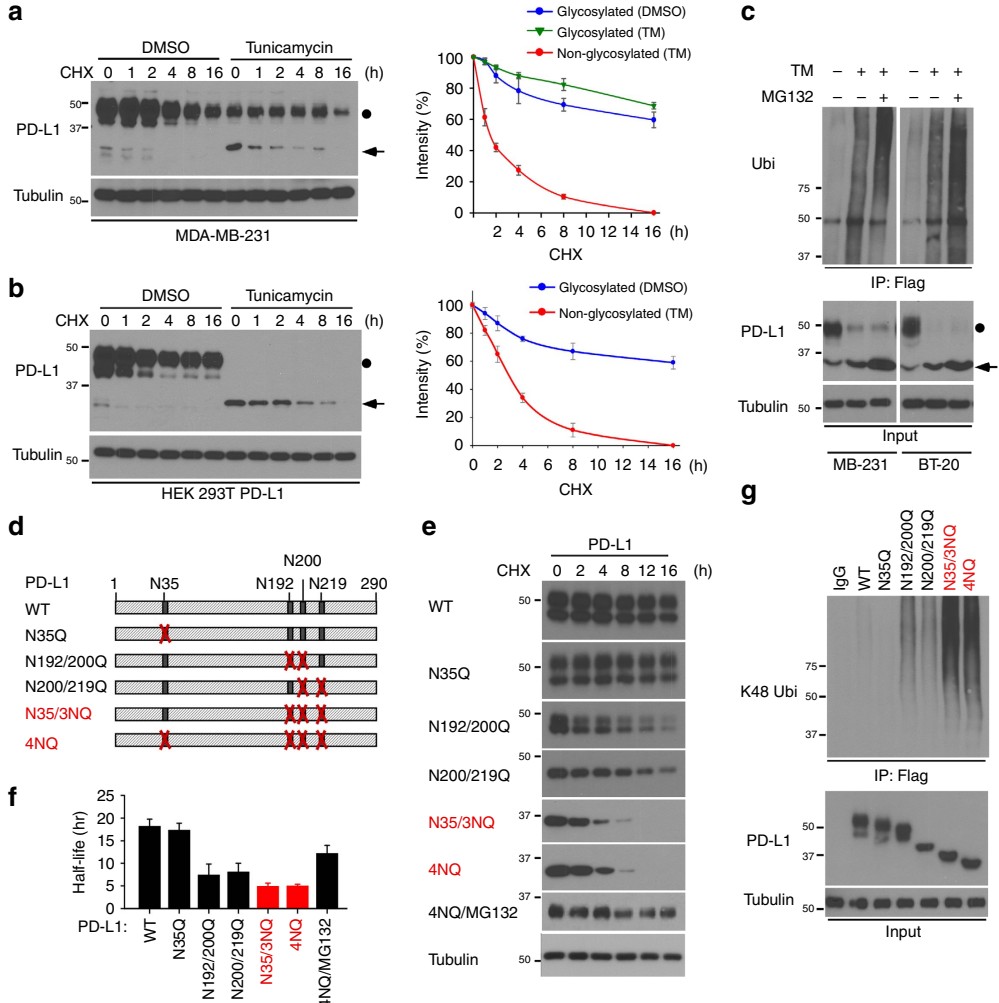

**Figure 2 | Glycosylation of PD-L1 stabilizes PD-L1 protein.** (**a**,**b**) Western blot analysis of PD-L1 protein in PD-L1-Flag expressing MDA-MB-231 cells (**a**) and HEK 293T cells (**b**). Cells were treated with 20 μM cycloheximide (CHX) at indicated intervals and analysed by western blot analysis. The intensity of PD-L1 protein was quantified using a densitometer. (**c**) Inhibition of PD-L1 glycosylation enhances ubiquitination. Breast cancer cells with TM and/or MG132 treatment were subjected to PD-L1 immunoprecipitation (IP) and western blot analyses with anti-K48 ubiquitin. (**d**) Schematic diagram of various PD-L1 NQ mutants used in this study. The numbers indicate amino acid positions on the PD-L1. (**e**) Protein half-life of PD-L1 WT or various NQ mutants expressing in MDA-MB-231 cells. Experiments were performed as described in (**a**). Quantification of PD-L1 half-life was shown in (**f**). (**g**) Ubiquitination of PD-L1 proteins in PD-L1 WT or various NQ mutants expressing MDA-MB-231 cells. PD-L1 proteins were IP with Flag antibody and then immunoblotted with ubiquitin antibody. Approximately 5% of the cell extract from IP was saved as input. Black circle, glycosylated PD-L1; arrowhead, non-glycosylated PD-L1. All error bars are expressed as mean ± s.d. of 3 independent experiments.

mediated phosphorylation has been shown to trigger interaction with E3 ligase and contribute to protein stability[19,20]. Thus, we first test the association between GSK3β and PD-L1. We used GST pull-down assay (Fig. 3b), co-immunoprecipitation assays (Supplementary Fig. 5c) and Duolink assay (Supplementary Fig. 5d), and found that PD-L1 4NQ but not WT interacts with GSK3β endogenously. In-depth analysis showed that GSK3β selectively bound to the C-terminal domain of non-glycosylated PD-L1 (Fig. 3c and Supplementary Fig. 5e). Deletion of the region (amino-acid residues 189–220) containing N192, N200 and N219 (PD-L1-Δ3N) disrupted its binding to GSK3β (Fig. 3c). Together, these results suggest that the region containing N192, N200 and N219 residues is required for PD-L1 binding to GSK3β, and glycosylation of N192, N200 and N219 may antagonize PD-L1 and GSK3β interaction.

To determine whether PD-L1 is a potential substrate of GSK3β, we conducted an immune-complex kinase assay. GST-PD-L1 was strongly phosphorylated by GSK3β CA (constitutively active,

GSK3β S9A) but not GSK3β kinase dead (KD, GSK3β S85A), and site-directed mutation of the consensus motif on PD-L1 (S176A, T180A and S184A) completely abolished PD-L1 phosphorylation (Fig. 3d). Among these three sites, we found that PD-L1 S176A is not involved in GSK3β-mediated PD-L1 phosphorylation, suggesting that T180 and S184 are responsible for PD-L1 phosphorylation by GSK3β (see later Fig. 3e). To pinpoint the phosphorylation ability according to the glycosylation state, PD-L1 WT and 4NQ were overexpressed together with GSK3β CA. The phospho-tag gel analysis revealed that GSK3β CA induced a noticeable mobility shift in PD-L1 4NQ but not in the WT. Moreover, in the presence of λ-phosphatase (Supplementary Fig. 5f), the mobility shift of 4NQ disappeared, suggesting that GSK3β phosphorylates the non-glycosylated PD-L1 but not WT PD-L1. To dissect GSK3β-mediated PD-L1 phosphorylation, we generated and validated the specificity of site-specific antibodies against PD-L1 T180 or S184 phosphorylation by which phosphopeptide but not non-phosphopeptide was

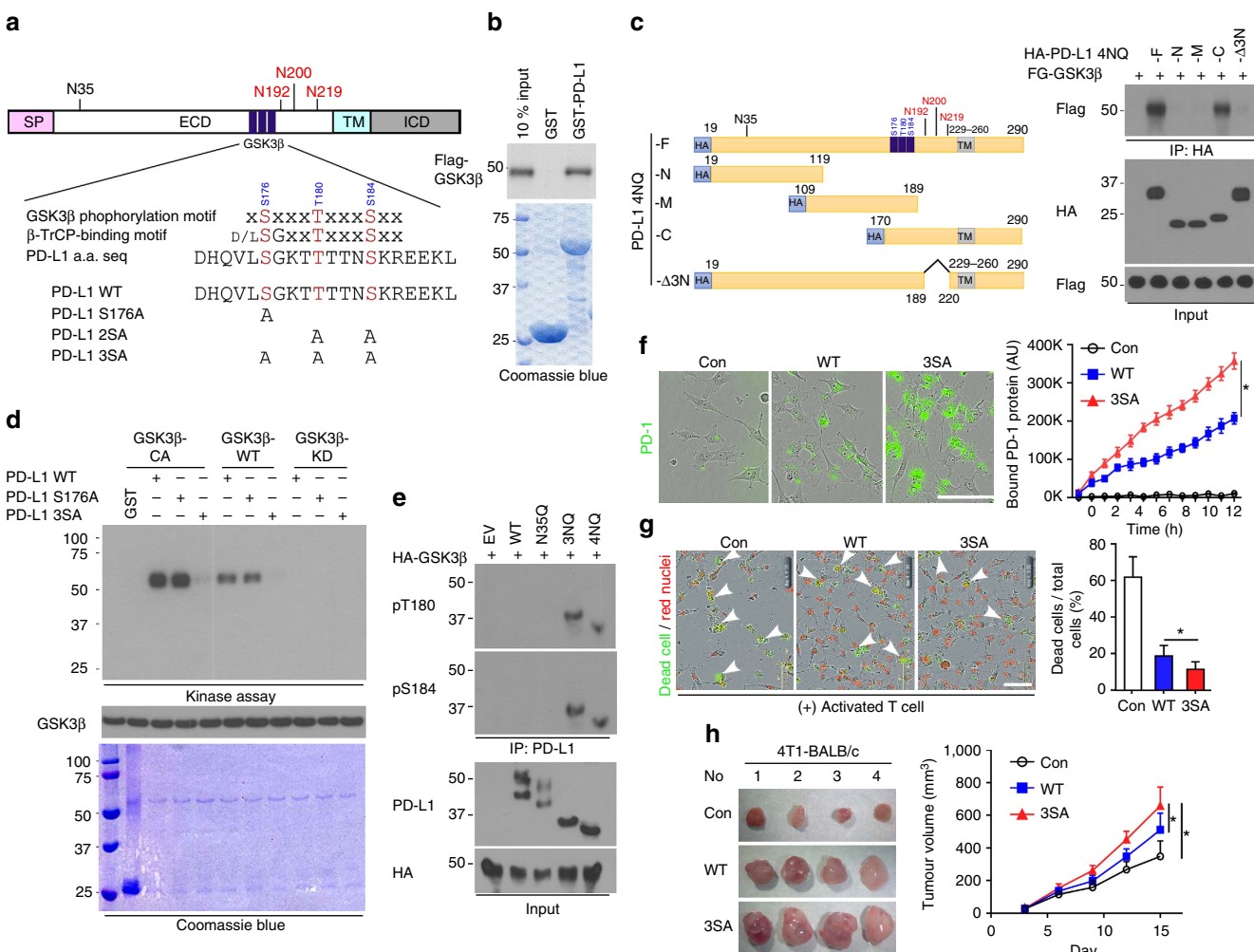

**Figure 3 | GSK3β binds to and phosphorylates PD-L1 protein *in vitro* and *in vivo*.** (**a**) Schematic diagram of GSK3β phosphorylation and β-TrCP-binding motifs and various mutants of PD-L1 expression constructs. PD-L1 was separated into ECD and ICD. SP, signal peptide; TM, transmembrane domain. The numbers indicate amino-acid positions. (**b**) *In vitro* GST pull-down assay of non-glycosylated PD-L1 and GSK3β. (**c**) Co-immunoprecipitation (co-IP) measuring the interaction of GSK3β and PD-L1 4NQ. Schematic diagram of PD-L1 4NQ deletion or truncation mutants showing on the left. Positions of glycosylation sites were labelled with red colour. The numbers indicate amino-acid positions. (**d**) Immunocomplex kinase assay measuring PD-L1 phosphorylation by GSK3β. Coomassie blue staining showing equal loading amount of GST-PD-L1. CA, constitutive activation mutant (S9A); KD, kinase dead (K85A) mutant; WT, wild type. (**e**) Western blot analysis of phosphorylation of PD-L1 protein at T180 and S184 sites by phospho-T180 and -S184 PD-L1 antibodies, respectively. EV, empty vector. (**f**) Time-lapse microscopy image (at 12 h) showing the dynamic interaction between PD-L1 and PD-1 at the last time point. The kinetic graph showed the quantitative binding of green fluorescent labelled PD-1/Fc protein on PD-L1 WT, 3SA or 4NQ expressing BT549 cells at every hour time point (right). Scale bar, 100 μm. (**g**) T-cell-mediated tumour cell-killing assay in PD-L1 WT or 3SA-expressing BT549 cells. Representative phase, red fluorescent (nuclear-restricted RFP), and/or green fluorescent (Caspase 3/7 substrate)-merged images of PD-L1 WT- or PD-L1 3SA-expressing cells and activated T-cell co-cultures at 96 h. Green fluorescent cell was counted as dead cell. The quantitative ratio of dead cells showed in bar graph (right). Scale bar, 100 μm. (**h**) The tumour growth of mouse PD-L1 WT- or PD-L1 3SA-expressing 4T1 cells in BALB/c mice. Quantification of tumour volume is shown on the right and representative images of tumours are shown on the left. *n* = 7 mice per group. Con, vector control; WT, PD-L1 WT; 3SA, PD-L1 3SA. **P* < 0.05 is statistically significant as shown by Student's *t*-test. All error bars are expressed as mean ± s.d. of three independent experiments.

recognized (Supplementary Fig. 5g). GSK3β CA-mediated PD-L1 T180 and S184 phosphorylation was observed only in PD-L1 3NQ and 4NQ but not in WT and N35Q (Fig. 3e), further supporting the notion that glycosylation of PD-L1 prevents GSK3β from binding to PD-L1 (Fig. 3c and Supplementary Fig. 5e). Together, the results suggest that GSK3β binds and phosphorylates the non-glycosylated PD-L1.

To determine whether GSK3β-mediated phosphorylation involves PD-L1 degradation, we assessed the E3 ligase and its physiological effects on PD-L1. We noted that PD-L1 contains a putative β-TrCP destruction box, D/LSGXXS, that is present next to GSK3β phosphorylation sites (Fig. 3a)[24,25]. Non-glycosylated

PD-L1 with mutation of this motif (LSGKTT to LAGKTT, PD-L1 S176A) resisted to GSK3β-mediated PD-L1 degradation (Supplementary Fig. 6a) as GSK3β failed to degrade PD-L1 4NQ S176A mutant. The degradation of PD-L1 is kinase-dependent because only GSK3β WT and CA GSK3β, but not KD GSK3β, induced degradation (Supplementary Fig. 6a). Similar to many β-TrCP substrates[24,25], induction of β-TrCP expression in a BT549-PD-L1 4NQ stable clone by ciglitazone, troglitazone or resveratrol[26,27] reduced PD-L1 expression (Supplementary Fig. 6b). Overexpression of both GSK3β and β-TrCP reduced PD-L1 4NQ (arrowhead) expression but not PD-L1 WT (black dot) when coexpressing PD-L1 WT and 4NQ together

in the assay (Supplementary Fig. 6c). Using six histidine-tagged ubiquitin to pull down substrates that covalently conjugated with ubiquitin, β-TrCP was found to catalyse PD-L1 ubiquitination in the presence of GSK3β and MG132 (Supplementary Fig. 6d). In contrast, deletion of the F-box in the β-TrCP or mutation of the GSK3β phosphorylation motif (PD-L1 2SA and 3SA, Fig. 3a) abrogated GSK3β-mediated PD-L1 ubiquitination, suggesting that ubiqiutin-E3 ligase activity is involved in PD-L1 stability (Supplementary Fig. 6e).

Since activation of GSK3β destabilizes PD-L1, which inhibits T-cell immunity, we hypothesized that GSK3β may regulate cancer immunosuppression via PD-L1 destabilization. To this end, GSK3β was stably knocked down using six independent shRNAs in MDA-MB-468 cells (Supplementary Fig. 7a), and Flag-tagged GSK3β was ectopically expressed in the No. 5 shRNA clone (Supplementary Fig. 7b, vector design). Restoration of Flag-tagged GSK3β WT and the CA form, but not KD in a low-GSK3β background, reduced PD-L1 expression (Supplementary Fig. 7c), PD-1 interaction (Supplementary Fig. 7d) and the immunosuppressive activity, as measured by increased interleukin (IL)-2 expression through co-culture with T cells (Supplementary Fig. 7e,f). In fact, the impact of GSK3β-mediated PD-L1 degradation can be found in both glycosylated and non-glycosylated PD-L1 as both PD-L1 3SA and PD-L1 4NQ/3SA exhibit better stability (Supplementary Fig. 7g) and lesser ubiquitination (Supplementary Fig. 7h) in both WT and 4NQ backgrounds. To determine whether GSK3β-mediated PD-L1 destabilization affects cancer cell immunosuppression, we compared the immunosuppression activity of PD-L1 WT and 3SA both in vitro and in vivo. Cells with PD-L1 3SA exhibited more PD-1 protein binding to the cell surface than did cells with PD-L1 WT (Fig. 3f). Consistently, the cells expressing PD-L1 3SA were more resistant to human T-cell-mediated cytolysis than were the cells with PD-L1 WT expression (Fig. 3g and Supplementary Fig. 7i,j, illustrated methodology). To verify this result in vivo, 4T1 cells stably expressing mouse PD-L1 WT and 3SA were inoculated to the mammary fat pad of BALB/c mice. The 4T1 tumours with PD-L1 3SA were more malignant (Fig. 3h) than those with PD-L1 WT. In addition, in tumour-infiltrating lymphocyte profile analysis, the population of activated cytotoxic T cells (CD8 and interferon gamma (IFNγ) positive) in 4T1 3SA tumours was lower than that in 4T1 WT tumours (Supplementary Fig. 7k). These results support the notion that stabilization of PD-L1 by inactivation of GSK3β enhances tumour-immunosuppressive function and gives an advantage for tumour cell survival in an in vivo mouse model.

**EGF signalling induces PD-L1 glycosylation.** To identify the upstream signalling that governs PD-L1 stabilization, we subjected various cancer cell lines to several growth factors that are known to inhibit GSK3β activity, such as epidermal growth factor (EGF), insulin-like growth factor-1, hepatocyte growth factor, fibroblast growth factor and transforming growth factor (TGF)-β. Among those examined, only EGF strongly induced PD-L1 expression in BT549 and MB-468 cells (Fig. 4a top, Fig. 4b,c and Supplementary Fig. 8a). Similarly, other EGFR ligands such as epiregulin, TGFα and heparin-binding EGF also induced PD-L1 expression (Fig. 4d). Unlike IFNγ-induced PD-L1 via transcription activation[2,4], the action of EGF-mediated PD-L1 induction is primarily at post-translational level as EGF did not influence PD-L1 mRNA expression (Supplementary Fig. 8b). Although both EGF and IFNγ induced endogenous PD-L1 at a similar level (Fig. 4e, lower panel), the exogenous PD-L1 (Flag-PD-L1, detected by Flag antibody), which was driven by a cytomegalovirus (CMV) promoter, was induced by EGF but not

by IFNγ (Fig. 4e, upper panel), further supporting the differential mechanisms of EGF and IFNγ to enhance PD-L1 expression. The pathological relevance of the identified mechanism was validated by the expression of p-EGFR (Tyr 1068), p-GSK3β (Ser 9), PD-L1 and the cytotoxic T-cell activation indicator granzyme B in human breast tumour specimens using immunohistochemical (IHC) staining in which PD-L1 expression correlated positively with p-EGFR ($P = 0.007$, Pearson $\chi^2$-test) and p-GSK3β ($P = 0.0001$, Pearson $\chi^2$-test) but negatively with granzyme B ($P = 0.043$, Pearson $\chi^2$-test; Supplementary Fig. 8c and Supplementary Table 1). Of note, the majority of samples with low PD-L1 had high p-EGFR expression. Therefore, EGFR-mediated PD-L1 stabilization may appear in a subset of EGFR-positive patients. Together, these data suggest that activation of EGFR may inactivate GSK3β and thereby stabilizes PD-L1 expression. The stabilized PD-L1 accounts for breast cancer cell immunosuppression (Supplementary Fig. 8d, proposed model).

**Gefitinib sensitizes PD-1 blockade therapy *in vivo*.** The proposed model prompted us to test a hypothesis whether inhibition of EGF-mediated PD-L1 stabilization may enhance blockage of the PD-L1/PD-1 therapy. To this end, we treated cells with an EGFR inhibitor (gefitinib, erlotinib, lapatinib or AG1478). Both EGF-induced PD-L1 expression (Fig. 5a) and ectopic PD-L1 expression (Fig. 5b) were significantly reduced in basal-like breast cancer (BLBC) cells. While PD-L1 is a well-known ligand of PD-l, PD-L1 also binds to CD80 for T-cell suppression[28,29]. To block PD-L1/PD-1 as well as other potential ligand/receptor binding such as PD-L1/CD80 effectively, we combined gefitinib and anti-PD-1 antibody for treatment. Because BLBC cells exhibit high levels of EGFR and PD-L1 (Supplementary Fig. 1a), we evaluated the combinatorial effect of gefitinib and anti-PD-1 antibody in BLBC cells. We found that gefitinib significantly increased the immune response of anti-PD-1 antibody by reducing the interaction between PD-L1 and PD-1 (Fig. 5c), by enhancing IL-2 expression in T cells (Fig. 5d), and by elevating T-cell-mediated tumour cell killing (Fig. 5e). Consistent with our observations *in vitro*, gefitinib enhanced anti-PD-1 antibody efficacy in a 4T1-luciferase (4T1-Luc) syngeneic BALB/c model. Tumour size was reduced, and mouse survival improved in both gefitinib- and anti-PD-1 antibody-treated mice (Fig. 5f–j), with no significant changes in body weight (Supplementary Fig. 9a) and minimal cytotoxicity in the liver and kidney (Supplementary Fig. 9b). The tumour-infiltrated activated CD8$^+$ T-cell population also significantly increased in both gefitinib- and anti-PD-1 antibody-treated mice (Fig. 5i,j and Supplementary Fig. 9c,d). In addition, gefitinib also enhanced an efficacy of anti-PD-1 antibody treatment in other syngeneic animal models such as EMT6 (mouse breast cancer cells) and CT26 (mouse colon cancer cells; Supplementary Fig. 9e–g). In fact, gefitinib provides multiple lines of benefits. It reduces PD-L1 expression and thus limits its binding to other T-cell receptors. Gefitinib may also limit PD-L1 oncogenic potential. Lastly, it reduces EGFR-overexpressing cell survivals. Together, inhibition of PD-L1 stabilization by gefitinib enhances an efficacy of PD-1 blockade in syngeneic mouse models.

**Discussion**
Unlike CTLA-4 or PD-1, which are primarily expressed in immune cells[14,30], PD-L1 is expressed in cancer cells and macrophages and plays a major role in inhibiting immune surveillance[4]. In this study, we dissected the mechanisms by which cancer cells initiate T-cell immunosuppression by inducing

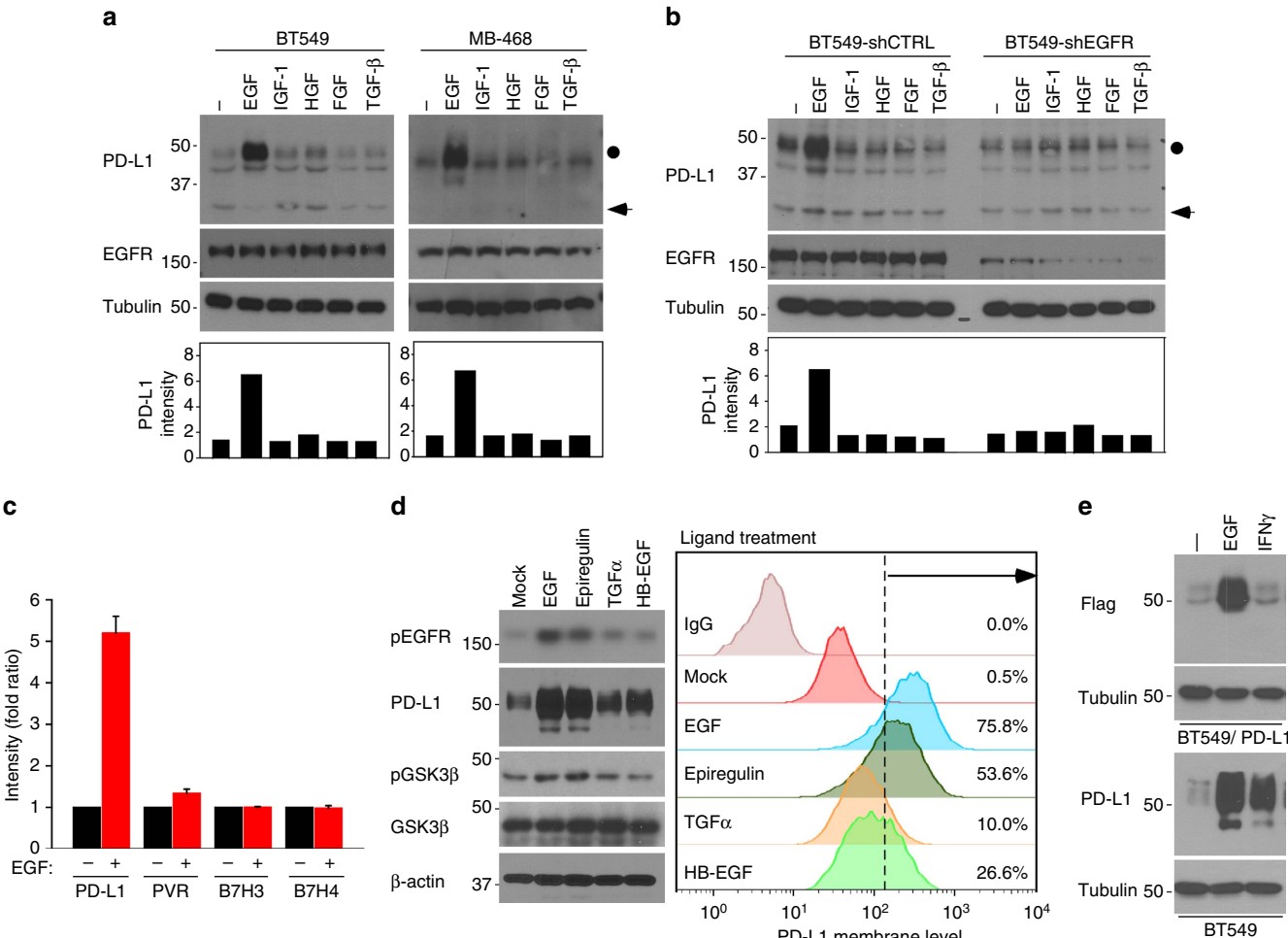

**Figure 4 | EGF signalling induces PD-L1 glycosylation. (a)** Western blot analysis of PD-L1 expression in BT549 and MB-468 cells treated with 25 ng ml$^{-1}$ EGF, 25 ng ml$^{-1}$ insulin-like growth factor-1, 10 ng ml$^{-1}$ hepatocyte growth factor, 25 ng ml$^{-1}$ fibroblast growth factor and 100 nM TGFβ for overnight. shCTRL, control shRNA. **(b)** Western blot analysis of PD-L1 expression in BT549-shCTRL and BT549-shEGFR cells. **(c)** Quantification of western blot results from Supplementary Fig. 8a. Cells were in a serum-free culture medium for overnight and then treated with EGF. The intensity of PD-L1 protein was quantified using a densitometer. **(d)** Western blot analysis of PD-L1 expression upon different agonist treatments for overnight. Cell surface analysis of PD-L1 protein using flow cytometer is shown in the right. **(e)** Flag-tagged PD-L1 was stably expressed in BT549 cells. Western blot analysis showing exogenous PD-L1 expression under EGF treatment.

PD-L1 stabilization. We first showed that glycosylation of PD-L1 inhibits 26S proteasome-mediated protein degradation. While dissecting the asparagine residue involved in such regulation, we discovered that N192, N200 and N219 are responsible for PD-L1 stabilization. In-depth analysis revealed GSK3β as a central node in regulating PD-L1 stability. Of note, single protein can be regulated by multiple kinases[31,32]. It is not yet clear whether other kinases may also phosphorylate PD-L1 to regulate its protein stability. If so, similar combination strategy may be explored to improve immune checkpoint therapy.

High AKT activation in cells with loss of PTEN has been shown to upregulate PD-L1 expression[33]. In addition, AKT activation by EGFR is correlated with membrane PD-L1 expression and poor survival in lung cancer patients[34]. Since activation of AKT by EGFR suppresses GSK3β activity through Ser9 phosphorylation[35], inhibition of EGF signalling in BLBC cells creates a therapeutic benefit by reducing cancer cell immune escape via PD-L1 destabilization. Consistent with the notion, PD-L1 is upregulated in mouse lung tumours carrying EGFR-activating mutations[34]. Thus far, we do not know whether AKT directly regulates PD-L1 expression. More

in-depth analysis would be required to justify the role of AKT in EGFR-mediated PD-L1 protein stabilization.

In summary, we demonstrated a novel interchange between glycosylation and phosphorylation regulating ubiquitination and degradation of PD-L1. This regulatory event is critical for BLBC cells that escape immune surveillance via PD-L1/PD-1 interaction. Importantly, inhibition of EGF-mediated PD-L1 stabilization enhances a therapeutic efficacy of PD-1 blockade to promote tumour-infiltrating cytotoxic T-cell immune response (Fig. 5i and Supplementary Fig. 9c–g). Thus, targeting PD-L1 stabilization provides a novel strategy to combat BLBC-mediated immunosuppression and may potentially apply to other cancer types.

## Methods

**Cell culture and transfection.** All cell lines were obtained from the American Type Culture Collection (Manassas, VA) and have been independently validated using STR DNA fingerprinting at MD Anderson, and tests for mycoplasma contamination were negative. These cells were grown in DMEM/F12 or RPMI 1640 medium supplemented with 10% fetal bovine serum. PD-L1-stable transfectants in MDA-MB-231, MDA-MB-468, BT549 and HEK 293T cells were selected using puromycin (InvivoGen, San Diego, CA, USA). For transient transfection, cells were

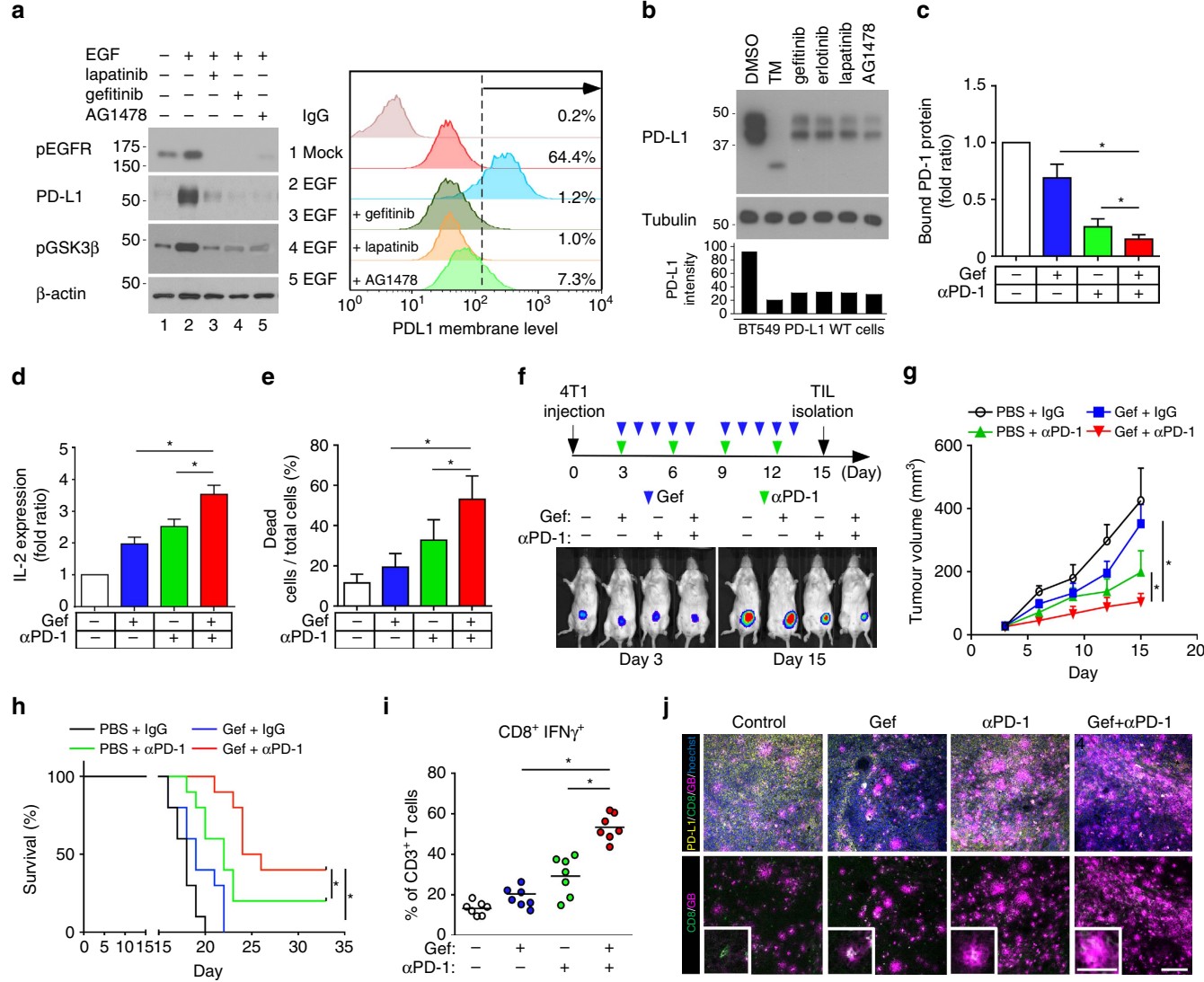

**Figure 5 | Inhibition of EGFR sensitizes the PD-1 blockade therapy in syngeneic mouse model.** (**a**) Cells were treated with TKIs for 2 h before EGF stimulation. Cell surface analysis of PD-L1 protein using flow cytometer was shown in the right. (**b**) Western blot analysis of PD-L1 protein in the cells treated with several indicated inhibitors. PD-L1 WT-expressing BT549 cells were treated with 1 μg ml$^{-1}$ TM, 1 μM gefitinib, 1 μM erlotinib, 1 μM lapatinib and AG1478. (**c**) PD-L1 and PD-1 interaction in PD-L1-expressing BT549 cells. (**d**) Soluble IL-2 levels in PD-L1-expressing BT549 cells treated with gefitinib and/or anti-PD-1 antibody. (**e**) T-cell-meditated killing of PD-L1-expressing BT549 cells treated with gefitinib and/or anti-PD-1 antibody. (**f**) The tumour growth of 4T1-Luc cells in BALB/c mice following treatment with gefitinib and/or anti-PD-1 antibody. Treatment protocol is summarized (top). Tumour growth of 4T1-Luc cells was shown *in vivo* by bioluminescence imaging using IVIS100 (bottom). (**g**) The tumour growth of 4T1 cells in gefitinib- and/or anti-PD-1 antibody-treated BALB/c mice. Tumours were measured at the indicated time points and dissected at end point. Quantification of tumour volume is shown on the right and representative images of tumours are shown on the left. $n = 9$ mice per group. (**h**) Survival of mice bearing syngeneic 4T1-Luc-derived tumour following treatment with gefitinib and/or anti-PD-1 antibody. Significance was determined by log-rank test. *$P < 0.05$; $n = 10$ mice per group. (**i**) Intracellular cytokine stain of IFNγ and CD8 in CD3$^+$ T-cell populations from the isolated tumour-infiltrating lymphocytes. (**j**) Immunofluorescence staining of the protein expression pattern of PD-L1, CD8 and granzyme B (GB) in 4T1 tumour mass. *$P < 0.05$ is statistically significant as shown by Student's *t*-test. All error bars are expressed as mean ± s.d. of three independent experiments. Gef, gefitinib.

transiently transfected with DNA using SN liposomes[36] and lipofectamine 2000 (Life Technologies, Carlsbad, CA, USA).

**Animal treatment protocol.** All BALB/C mouse (6–8-week-old female; Jackson Laboratories) procedures were conducted under the guidelines approved by the Institutional Animal Care and Use Committee at MD Anderson Cancer Center. Mice were divided according to the mean value of tumour volume in each group. 4T1-Luc cells ($5 \times 10^4$ cells in 50 μl medium mixed with 50 μl Matrixgel Basement Membrane Matrix (BD Biosciences, San Jose, CA, USA) were injected into the mammary fat fad. For treatment with antibody, 100 μg anti-PD-1 antibody (RMP1-14, Bio X Cell, West Lebanon, NH, USA) or rat IgG (Bio X Cell) as control was injected intraperitoneally on days 3, 6, 9 and 12 after 4T1 cell inoculation. For drug treatment, mice were treated with daily oral doses of 10 mg kg$^{-1}$ gefitinib for 2 weeks (5 days per week). Tumour was measured

weekly with a caliper, and tumour volume was calculated using the formula: $\pi/6 \times \text{length} \times \text{width}^2$.

**Tumour infiltration lymphocyte profile analysis.** Mice receiving a $5 \times 10^4$ 4T1-Luc cell challenge were treated with anti-PD-1 and/or gefitinib as discussed in the previous section. Excised tumours were digested in collagenase/hyalurinidase (Stemcell Technologies, Vancouver, BC, Canada) and DNase (Sigma), and lymphocytes were enriched on a Ficoll gradient (Sigma) and then T cells were isolated using Dynabeads untouched mouse T-cell kit (Invitrogen). T cells were stained using CD3-PerCP (BioLegend, San Diego, CA, USA), CD4-FITC (eBioscience, San Diego, CA, USA), CD8-APC/Cy7 (BioLegend), CD45.1-PE (BioLegend) and IFNγ-Pacific Blue antibodies. Stained samples were analysed using BD FACSCanto II (BD Biosciences) cytometer.

**Generation of stable cells using lentiviral infection.** The lentiviral-based shRNA (pGIPZ plasmids) used to knock down expression of human or mouse PD-L1 (ref. 37) was purchased from the shRNA/ORF Core Facility (UT MD Anderson Cancer Center). On the basis of knockdown efficiency of PD-L1 protein expression in MDA-MB-231 or A431 cells, we selected two shPD-L1 clones for this study. The mature antisense sequences are as follows: 5′-TCAATTGTCATATTGCTA C-3′ (shPD-L1 #1), 5′-TTGACTCCATCTTTCTTCA-3′ (shPD-L1 #5). Using a pGIPZ-shPD-L1/Flag-PD-L1 dual expression construct to knock down endogenous PD-L1 and reconstitute Flag-PD-L1 simultaneously, we established endogenous PD-L1 knockdown and Flag-PD-L1 WT, 3SA, or 4NQ mutants expressing cell lines. To generate lentivirus-expressing shRNA for PD-L1 and Flag-PD-L1, we transfected HEK 293T cells with pGIPZ-non-silence (for vector control virus), pGIPZ-shPD-L1 or pGIPZ-shPD-L1/PD-L1 WT, pGIPZ-shPD-L1/PD-L1 4NQ mutant or pGIPZ-shPD-L1/PD-L1 3SA mutant with FuGENE 6 transfection reagent. Twenty-four hours after transfection, the medium was changed, and then it was collected at 24-h intervals. The collected medium containing lentivirus was centrifuged to eliminate cell debris, and was filtered through 0.45-µm filters. Cells were seeded at 50% confluence 12 h before infection, and the media were replaced with a medium containing lentivirus. After infection for 24 h, the medium was replaced with fresh medium and the infected cells were selected with 1 µg ml$^{-1}$ puromycin (InvivoGen).

**Plasmids.** The human PD-L1 clone was obtained from the shRNA/ORF Core Facility (UT MD Anderson Cancer Center, Houston, TX, USA) and cloned into pCDH lentiviral expression vectors to establish PD-L1-Flag or PD-L1-Myc expression cell lines. In addition, it also cloned into pEGFP-N1 and pCMV-HA mammalian cell expression vectors for transient transfection. Using the pCDH/PD-L1-Flag expression vector as a template, we made PD-L1-Flag NQ mutants (N35Q, N192Q, N200Q, N219Q), and 4NQ (N35Q/N192Q/N200Q/N219Q) were developed by performing a site-directed mutagenesis (Supplementary Table 1). To create a pGIPZ-shPD-L1/Flag-PD-L1 dual expression construct to knock down endogenous PD-L1 and reconstitute Flag-PD-L1 simultaneously, first, we selected a shPD-L1 construct (shPD-L1 #5) that targets the 3′-untranslated repeat region of PD-L1 mRNA. And then Flag-PD-L1 WT or 4NQ mutant were cloned into pGIPZ-shPD-L1 (Thermo Scientific, Pittsburgh, PA, USA), expressing shRNA for endogenous PD-L1. To create a pGIPZ-shmPD-L1/Flag-mPD-L1 (mouse PD-L1) or a pGIPZ-shGSK3β/Flag-GSK3β dual expression construct, we used the same method as the pGIPZ-shPD-L1/Flag-PD-L1 dual expression construct. All constructs were confirmed using enzyme digestion and DNA sequencing. Detailed information is available upon request.

**qRT–PCR assays.** Quantitative reverse transcriptase PCR (qRT–PCR) assays were performed to measure the expression of mRNA[37,38]. Cells were washed twice with PBS and immediately lysed in QIAzol. The lysed sample was subjected to total RNA extraction using the RNeasy Mini Kit (Qiagen, Hilden, Germany). To measure the expression of mRNA, cDNA was synthesized from 1 µg purified total RNA with the SuperScript III First-Strand cDNA synthesis system using random hexamers (Life Technologies) according to the manufacturer's instructions. qPCR was performed using a real-time PCR machine (iQ5, Bio-Rad, Hercules, CA, USA) with the following primers: 5′-TCACTTGGTAATTCTGGGAGC-3′ (PD-L1 forward), 5′-C TTTGAGTTTGTATCTTGGATGCC-3′ (PD-L1 reverse), 5′-GCAAAGACCTGT ACGCCAACA-3′ (β-actin reverse) and 5′-TGCATCCTGTCGGCAATG-3′ (β-actin reverse). All the data analyses were performed using the comparative Ct method. Results were first normalized to internal control β-actin mRNA.

**Antibodies and chemicals.** The following antibodies were used: Flag (F3165; Sigma-Aldrich, St Louis, MO, USA; 1:5,000), Myc (11667203001; Roche Diagnostics, Indianapolis, IN, USA; 1:5,000), HA (11666606001; Roche Diagnostics; 1:5,000), PD-L1 (13684; Cell Signaling Technology, Danvers, MA, USA; 1:1,000), PD-L1 (329702; BioLegend; 1:1,000), PD-L1 (GTX117446; GeneTex, Irvine, CA, USA; 1:1,000), PD-L1 (AF156; R&D Systems, Minneapolis, MN, USA; 1:1,000), PD-1 (ab52587; Abcam, Cambridge, MA, USA; 1:2,000), Granzyme B (ab4059; Abcam; 1:1,000), EGFR (4267; Cell Signaling Technology; 1:1,000), GSK3β (9315; Cell Signaling Technology; 1:1,000), phospho-GSK3β Ser 9 (9336; Cell Signaling Technology; 1:1,000), β-TrCP (4394; Cell Signaling Technology; 1:1,000), α-Tubulin (B-5-1-2; Sigma-Aldrich; 1:5,000), β-Actin (A2228; Sigma-Aldrich; 1:10,000). EGF, cycloheximide and TM were purchased from Sigma-Aldrich. Gefitinib, erlotinib, lapatinib, cetuximab and AG1478 were obtained from Calbiochem Corp (Billerica, MA, USA).

**Antibody generation.** The mouse anti-phospho-PD-L1 T180 antibody was raised against a phosphorylated synthetic peptide (C-QVLSGKTT(p)TTNSKREE-NH2), and the mouse anti-phospho-PD-L1 S184 antibody was generated from a phosphorylated synthetic peptide (C-GKTTTTNS(p)KREEKLF-NH2). Antibodies were generated as previously described[39]. Briefly, antibodies against T180 and S184 phosphorylation sites of PD-L1 were generated in LifeTein (Somerset, NJ, USA). Synthetic phosphorylated peptides representing portions of PD-L1 around either T180 or S184 were used as antigens for producing antibodies. The antibodies were then purified using a phosphopeptide column.

**Western blot analysis and immunoprecipitation.** Western blot analysis was performed as described previously[39,40]. Image acquisition and quantitation of band intensity were performed using Odyssey infrared imaging system (LI-COR Biosciences, Lincoln, NE, USA). For immunoprecipitation, the cells were lysed in buffer (50 mM Tris·HCl, pH 8.0, 150 mM NaCl, 5 mM EDTA and 0.5% Nonidet P-40) and centrifuged at 16,000g for 30 min to remove debris. Cleared lysates were subjected to immunoprecipitation with antibodies. To measure 2-DG incorporation on PD-L1 protein, cells were incubated with IRDye 800CW 2-DG Optical probe (LI-COR Biosciences) for overnight, and then we performed immunoprecipitation. Uncropped scans of the most important western blots are shown in Supplementary Fig. 10.

**Immunocytochemistry.** For immunocytochemistry, cells were fixed in 4% paraformaldehyde at room temperature for 15 min, permeabilized in 5% Triton X-100 for 5 min and then stained using primary antibodies. The secondary antibodies used were anti-mouse Alexa Fluor 488 or 594 dye conjugate and/or anti-rabbit Alexa Fluor 488 or 594 dye conjugate (Life Technologies). Nuclei were stained with 4′,6-diamidino-2-phenylindole (DAPI; blue; Life Technologies). After mounting, the cells were visualized using a multiphoton confocal laser-scanning microscope (Carl Zeiss, Thornwood, NY, USA).

**PD-L1 and PD-1 interaction assay.** To measure PD-1 and PD-L1 protein interaction, cells were fixed in 4% paraformaldehyde at room temperature for 15 min and then incubated with recombinant human PD-1 Fc protein (R&D Systems) for 1 h. The secondary antibodies used were anti-human Alexa Fluor 488 dye conjugate (Life Technologies). Nuclei were stained with DAPI (blue; Life Technologies). And then we measured the fluorescence intensity of Alexa Fluor 488 dye using a microplate reader Synergy Neo (BioTeK, Winooski, VT, USA) and normalized to the intensity by total protein quantity. To take an image, after mounting, the cells were visualized using a confocal laser-scanning microscope (Carl Zeiss). To monitor a dynamic PD-1 protein binding on live cell surface, PD-L1 WT or 3SA-expressing BT549 cells were incubated with Alexa Fluor 488 dye conjugate PD-1 Fc protein and taken a time-lapse image at every hour using the IncuCyte Zoom microscope (Essen Bioscience).

**T-cell-mediated tumour cell-killing assay.** T-cell-mediated tumour-cell-killing assay was performed according to the manufacturer's protocol (Essen Bioscience). To analyse the killing of tumour cells by T-cell inactivation, nuclear-restricted red fluorescent protein (RFP)-expressing tumour cells were co-cultured with activated primary human T cells (Stemcell Technologies) in the presence of caspase 3/7 substrate (Essen Bioscience). T cells were activated by incubation with anti-CD3 antibody (100 ng ml$^{-1}$) and IL-2 (10 ng ml$^{-1}$). After 96 h, RFP and green fluorescent (NucView 488 Caspase 3/7 substrate) signals were measured. Green-fluorescent cells were counted as dead cells.

**Co-culture experiments and IL-2 expression measurement.** Co-culture of Jurkat T cells and tumour cells and IL-2 expression measurement was performed as described previously[41]. To analyse the effect of tumour cells on T-cell inactivation, tumour cells were co-cultured with activated Jurkat T cells expressing human PD-1, which were activated with Dynabeads Human T-Activator CD3/CD28 (Life Technologies). Co-cultures at 5:1 (Jurkat: tumour cell) ratio were incubated for 12 or 24 h. Secreted IL-2 level in the medium was measured as described by the manufacturer (Human IL-2 ELISA Kits, Thermo Scientific).

**Glycosylation analysis of PD-L1.** To confirm glycosylation of PD-L1 protein, we treated the cell lysates with PNGase F, Endo H and O-glycosidase (New England BioLabs, Ipswich, MA, USA) as described by the manufacturer. To stain glycosylated PD-L1 protein, we stained purified PD-L1 protein using the Glycoprotein Staining Kit (Pierce/Thermo Scientific) as described by the manufacturer.

**IHC staining of human breast tumour tissue samples.** Human breast tumour tissue specimens were obtained following the guidelines approved by the Institutional Review Board at MD Anderson Cancer Center, and written informed consent was obtained from patients in all cases at the time of enrolment. IHC staining was performed as described previously[39,42]. Briefly, tissue specimens were incubated with antibodies against PD-L1 (Cell Signaling Technology, #13684, 1:30 dilution), p-EGFR (Y1068; #3777, Cell Signaling Technology, 1:200 dilution), p-GSK3β (S9; #9323, Cell Signaling Technology, 1:200 dilution) or Granzyme B (ab4059, Abcam, 1:100 dilution) and a biotin-conjugated secondary antibody and then incubated with an avidin–biotin–peroxidase complex. Visualization was performed using amino-ethylcarbazole chromogen. For statistical analysis, Fisher's exact test and Pearson $\chi^2$-test were used and a P value less than 0.05 is considered statistically significant. The intensity of staining was ranked into four groups: high (+ + +), medium (+ +), low (+) and negative (−) according to histological scores.

**Identification of N-glycopeptide.** Purified Flag-tagged PD-L1 protein was reduced by 10 mM dithiothreitol for 1 h at 37 °C, and then alkylated by 50 mM iodoacetamide for 1 h in the dark at room temperature, both in 25 mM ammonium bicarbonate buffer. Reduced and alkylated PD-L1 was then treated with sequencing grade trypsin (1:50, enzyme-to-substrate) at 37 °C overnight, after which the digested products were diluted with formic acid to a final concentration of 0.1% and desalted with ZipTip C18 (Millipore) before nano LC-MS/MS analysis on an Orbitrap Fusion Tribrid fitted with an UltiMate 3000 RSLCnano System (Thermo Scientific). Briefly, the peptide mixture was loaded on an Acclaim PepMap 100 (2 cm × 100 μm i.d; Dionex) trap column and then separated on an Acclaim PepMap RSLC 25 cm × 75 μm i.d. column (Dionex) by a gradient of 5–35% acetonitrile/0.1% formic acid in 60 min at a flow rate of 500 nl min$^{-1}$. MS and MS/MS data were acquired under the higher-energy collisional induced dissociation (HCD) product ion trigger collision-induced dissociation (CID) mode, using a 3-s top-speed-mode cycle time. Collision energy used for HCD (fourier transform mass spectrometry (FTMSn)) was 28% stepped at 5% and the product ions $m/z$ 204.0867 or 366.1396 were used to trigger CID (ion trap mass spectrometry (ITMSn)) at 30% collision energy. The raw data obtained were then converted to the Mascot generic format using Proteome Discoverer 1.4, and the resulting HCD MS$^2$ data were searched for tentative glycopeptide matches using Byonic v. 2.2.9 (Protein Metrics) with the following parameters: peptide tolerance, 2 p.p.m.; fragment tolerance, 6 p.p.m.; missed cleavages, 1; modifications, carbamidomethyl cysteine (fixed), methionine oxidation (common2) and deamidation at N (rare 1). The returned positive glycopeptide hits were then further validated manually by considering both HCD and CID MS$^2$ results.

**Statistical analysis.** Data in bar graphs represent the mean fold change relative to untreated or control groups with s.d. of three independent experiments. Statistical analyses were performed using SPSS (Ver. 20, SPSS, Chicago, IL). The correlation between protein expression was analysed using Pearson $\chi^2$-test and Mann–Whitney test. Student's $t$-test was performed for experimental data. A $P$ value < 0.05 was considered statistically significant.

**Data availability.** All data supporting the findings of this study are available with the article, or from the corresponding author upon reasonable request.

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

## Acknowledgements

This work was funded in part by the following: National Institutes of Health (grants R01 CA109311, PO1 CA099031 and CCSG CA016672); National Breast Cancer Foundation Inc., Breast Cancer Research Foundation (to M.-C.H. and G.N.H.); Patel Memorial Breast Cancer Endowment Fund; The University of Texas MD Anderson-China Medical University and Hospital Sister Institution Fund (to M.-C.H.); Ministry of Science and Technology, International Research-intensive Centers of Excellence in Taiwan (I-RiCE; MOST105-2911-I-002-302); Ministry of Health and Welfare, China Medical University Hospital Cancer Research Center of Excellence (MOHW105-TDU-B-212-134003); Center for Biological Pathways; Susan G. Komen for the Cure Postdoctoral Fellowship (PDF12231298 to S.-O.L.); Basic Science Research Program through the National Research Foundation of Korea funded by the Korea government (MSIP; NRF-2011-357-C00140 to S.-O.L.); the National Research Foundation of Korea (NRF) grant for the Global Core Research Center (GCRC) funded by the Korea government (MSIP; 2011-0030001 to J.-H.C.).

## Author contributions

C.-W.L. and S.-O.L. designed and performed the experiments, analysed data and wrote the manuscript; W.X., H.-H.L., C.-W.K., K.-H.K., S.-S.C., J.-M.H., J.Y., Q.D. and Y. Wang performed experiments and analysed data; C.-C.L. and H.-J.W. analysed data; Y. Wu provided patient tissue samples; J.L.H. and H.Y. provided scientific input and wrote the manuscript; A.A.S., D.Y. and G.N.H. provided scientific and clinical input. M.-C.H. supervised the entire project, designed the experiments, analysed data and wrote the manuscript.

## Additional information

**Competing financial interests:** M.-C.H. received sponsored research agreement from STCube Pharmaceuticals Inc. through MD Anderson Cancer Center. C.-W.L., S.-O.L. and M.-C.H. are inventors on patent applications under review: Dual function antibodies specific to glycosylated PD-L1 and methods of use thereof, 2016, No. 62/314,652. Combination treatments directed toward programmed death ligand-1 (PD-LI) positive cancers, 2016, No. 62/316,178. Antibodies specific to glycosylated PD-L1 and methods of use thereof, 2016, No. PCT/US16/24691. The remaining authors declare no competing financial interests.

**How to cite this article**: Li, C.-W. *et al.* Glycosylation and stabilization of programmed death ligand-1 suppresses T-cell activity. *Nat. Commun.* 7:12632 doi: 10.1038/ncomms12632 (2016).

