## [Peer Review File · Nature Communications]

Reviewers' Comments:

Reviewer #1 (Remarks to the Author)

This study shows that EGF signaling could stabilize PD-L1 expression on tumor cells, possibly via modulation of ubiquitination and N-glycosylation. PD-L1 (B7-H1) is the important target in cancer therapy at the present time. It was reported that mutations of EGFR in cancer (such as lung cancer) could upregulate PD-L1 and these mutations in EGFR lead to constitutive activation of EGFR signaling. Therefore, the study presented here is an extension of previous study but provides biochemical perspectives. These findings are interesting and important.

Specific comments:

1) PD-1 in cancer appears to be highly heterogenic with the major band in ~45KDa while there is also ~37KDa band (figure 1a, 1b). The authors speculate this heterogeneity is due to glycosylation. Does different glycosylations affect PD-L1's affinity to bind PD-1?

2) In figure 5g and 5h, combination of anti-PD-1 and gefitinib induced an additive antitumor effect. I do not understand this data. If anti-PD-1 antibody was used in the saturated amount in vivo (judging from what they injected), addition of gefitinib should not have any effect if the effect of EGF is to stabilize PD-L1 as they proposed. Gefitinib and anti-PD-1 works under the same general mechanism; one de-stablizing PD-L1 expression and another preventing PD-L1/PD-1 interaction. The net result is decreased interaction of PD-1/PD-L1 and anti-PD-1 has done it all.

Reviewer #2 (Remarks to the Author)

In this manuscript the authors have carried out experiments which suggest that the immunosuppression activity of PD-L1 is stringently modulated by ubiquitination and N-glycosylation. GSK-3 is also suggested to be a novel protein that interacts with PD-L1 and can induce phosphorylation-dependent proteasome degradation by E-TrCP. The hypothesis is interesting and somewhat unusual. To my knowledge, this is a "novel" observation although the interplay between ubiquitination and N-Glycosylation is well established. The authors have used a combination of site-directed mutagenesis and pharmacological probes to make their arguments. Specific comments follow:

The quantitation of gels in Figures 1-4 would greatly help the reader understand the quantitative changes. Really hard to decipher what his there now.

The data suggesting GSK-3 as the sole activator is lacking. This can only really be addressed by kinase array technology

The references seem appropriate

The model given in Fig 4 seems completely unnecessary since these are straightforward arguments.

The antitumor activity measurements are quite preliminary and may not warrant the sweeping generalizations

In summary, this is an interesting, if flawed contribution which could contribute to our understanding of how to better enhance cancer immunotherapy.

Reviewer #3 (Remarks to the Author)

In this paper, Li et al. examine the molecular and biochemical mechanisms that influence the stability of PD-L1 and how that impacts the check point function of this protein. The authors have convincingly shown that PD-L1 exists in a glycosylated and non-glycosylated form, they identified the sites on the molecule that are N-glycosylated, and they showed that glycosylation has a profound effect on protein stability, with the non-glycosylated version being subjected to phosphorylation and subsequent ubiquitylation and degradation. Based on these observations, the authors propose that growth factor signaling in cancer cells represses the activity of GSK3B, the kinase that phosphorylates PD-L1 resulting in glycosylation, stabilization and immune suppression. While this is an elegant model, their data do not agree with the prediction from the model which predicts that blocking growth factor signaling can reduce the immunosuppressive function of PD-L1.

The majority of the work presented in this paper was exceptionally well done and the data quality are outstanding in most of the figures. In some cases, appropriate statistical methods have been used, but this is not always the case. This is a very interesting and important paper, however, the authors have over-interpreted some of their results. Specific comments are below:

1. I don't understand figure 1d. I understand what the authors are trying to show, but this figure is unclear to me. By contrast, supplementary figure 1e is quite clear so I don't understand why this isn't the figure presented in the main part of the paper.
2. On page 6, the authors state "the endogenous PD-L1 turnover rate was much faster in TM treated cells than in DMSO treated cells." This needs to be more clearly stated as the figure shows that the turnover rate of only the non-glycosylated form had the faster turnover rate.
3. Next the authors state that MG132 increased non-glycosylated PD-L1 and its ubiquitination, however, this is not exactly what figure 2C shows, as there is no MG132 alone group, only cells treated with MG132 plus TM. So, that lane either needs to be added, or this sentence needs to be corrected.
4. Figure 2e shows the influence of removing glycosylation sites on PD-L1 and the effect of that on stability in the presence of CHX. It's not clear, however, why in the top three panels, two bands are shown while in the bottom four panels, only a single band is visible, and there are no markers to indicate which band we're looking at.
5. Figure 3B is entirely unclear. It is also unclear why the authors chose to show the results of a Duolink assay (I don't know what that is) instead of the more straightforward co-IPs and GST-pulldown assays that are shown in the supplement, to make this point. Once again, it seems that the supplementary data is much clearer than the data shown in the paper. Seeing dots on an IF is not very compelling. I suggest switching these out.
6. The authors point to figure 3c when they say that deletion of a region disrupts binding to GSK3B, but 3c is just a cartoon and the real data are again shown in the supplement. I recommend switching these too.
7. I am having trouble seeing the data in figure S6a on the role of the mutation in the betaTrCP destruction box and its effect on PD-L1 stability
8. The part of the paper that centers around the influence of signaling molecules and receptors on the stability of PD-L1 is the weakest part of the paper. First, it never inspires confidence when investigators keep switching cell lines in order to make specific points. The results with BT549 cells stimulated with EGF are interesting, but the A431 data are not meaningful because EGF induces profound apoptosis in this EGFR amplified cell line. Earlier in the paper, MDA-468 cells were used, which are also EGFR amplified. Why are data with that cell line not presented in this figure? Regarding figure 4b, which is connected to figure S8a, what is PVR, B7H3 and B7H4 and why are they in these figures? Furthermore, EGF did not have a consistent effect on PD-L1 levels in the other cell lines shown in figure S8a and in figure S8d, there is clearly no correlation between steady state EGFR levels and PD-L1 levels. Also, figure 4f has a model in which AKT is depicted,

but there are no AKT data in this paper.

9. Figure 4e is entirely unconvincing and there is supplementary Table 1 is not interpretable as there is not enough information provided about how these data were generated, nor how the statistical analysis was done.

10. Next the authors try to make the case that the interaction between EGFR signaling and PD-L1 stability has therapeutic indications, but here again; the connection is tenuous and not convincing. The authors claim a synergistic interaction between gefitinib treatment and anti-PD1 treatment in the 4T1 tumor model. However, there is no statistical indication for synergy in the data shown. Indeed, the data shown in figure 5g shows and effect of gefitinib alone, and anti-PD1 alone on tumor growth, and the combination treatment appears to have yielded an additive response, which is not consistent with an interaction between the two drugs. In this regard, the data in figure 5i is more convincing and suggestive on an enhanced immune response following the combined treatment.

11. Because the association of PD-L1 stability with EGFR signaling is not convincing, the title of the paper needs to be changed to more accurately reflect the key findings of this paper.

In summary, this is an excellent paper on an important topic. The biochemistry is outstanding, but more work needs to be done to extend their model to tumor biology and immune therapy.

Point-by-Point Response Letter (NCOMMS-16-00937A)

Reviewers' comments:

Reviewer #1 (PD1 and Cancer Immune escape mechanisms) (Remarks to the Author):

This study shows that EGF signaling could stabilize PD-L1 expression on tumor cells, possibly via modulation of ubiquitination and N-glycosylation. PD-L1 (B7-H1) is the important target in cancer therapy at the present time. It was reported that mutations of EGFR in cancer (such as lung cancer) could upregulate PD-L1 and these mutations in EGFR lead to constitutive activation of EGFR signaling. Therefore, the study presented here is an extension of previous study but provides biochemical perspectives. These findings are interesting and important.

Specific comments:

1) PD-1 in cancer appears to be highly heterogenic with the major band in ~45KDa while there is also ~37KDa band (figure 1a, 1b). The authors speculate this heterogeneity is due to glycosylation. Does different glycosylations affect PD-L1's affinity to bind PD-1?

2) In figure 5g and 5h, combination of anti-PD-1 and gefitinib induced an additive antitumor effect. I do not understand this data. If anti-PD-1 antibody was used in the saturated amount in vivo (judging from what they injected), addition of gefitinib should not have any effect if the effect of EGF is to stabilize PD-L1 as they proposed. Gefitinib and anti-PD-1 works under the same general mechanism; one de-stablizing PD-L1 expression and another preventing PD-L1/PD-1 interaction. The net result is decreased interaction of PD-1/PD-L1 and anti-PD-1 has done it all.

Author's Response: We are deeply appreciative of the reviewer's comments and suggestions. Below please find our response to these comments in a point-by-point manner.

Redacted

Response to Point #2: We thank the reviewer for the comment. We have revised our manuscript to strengthen the rationale of combination strategy. While anti-PD-1 completely blocks the interactions such as those between PD-1/PD-L1 and PD-1/PD-L2, it does not completely block PD-L1-mediated T cell suppression as PD-L1 may induce T cell suppression via T cell receptors other than PD-L1. For example, PD-L1 also binds to CD80 and suppresses T cell activity^{1,2}. Other unidentified T cell receptors may exist and contribute to T cell suppression. In this sense, anti-PD-1 only provide partial blockade to shut off PD-1/PD-L1 axis. It is reasonable that TKI provide extra benefit to anti-PD-1 by downregulating PD-L1 expression.

In addition, it has been reported that PD-L1 harbors oncogenic potentials in cancer cells³ although the detailed mechanism is not yet completely understood. Furthermore, EGFR TKI, gefitinib itself can inhibit EGFR's survival signaling and has been used to treat patients with EGFR overexpression/activation cancers. Thus, we performed the combination therapy and indeed observed better therapeutic effect (Figure 5g-h).

On page 11 of the revised version, we added the following "*PD-1 is a well-known immune receptor of PD-L1 but PD-L1 also binds to CD80 for T cell suppression^{1,2}. To block PD-L1/PD1 as well as other possible ligand/receptor binding such as PD-L1/CD80 axis effectively, we combined gefitinib and anti-PD-1 antibody treatments.*"

We also added in the discussion "*In fact, TKI provides multiple lines of benefits. It reduces PD-L1 expression and thus limits its binding to other T cell receptors. TKI may also limit PD-L1 oncogenic potential. Lastly, it reduces EGFR overexpressing TNBC cell survivals*"

Reviewer #2 (Protein Glycosylation) (Remarks to the Author):

In this manuscript the authors have carried out experiments which suggest that the immunosuppression activity of PD-L1 is

stringently modulated by ubiquitination and N-glycosylation. GSK-3 is also suggested to be a novel protein that interacts with PD-L1 and can induce phosphorylation-dependent proteasome degradation by E-TrCP. The hypothesis is interesting and somewhat unusual. To my knowledge, this is a "novel" observation although the interplay between ubiquitination and N-Glycosylation is well established. The authors have used a combination of site-directed mutagenesis and pharmacological probes to make their arguments.

Specific comments follow:

The quantitation of gels in Figures 1-4 would greatly help the reader understand the quantitative changes. Really hard to decipher what his there now.

The data suggesting GSK-3 as the sole activator is lacking. This can only really be addressed by kinase array technology

The references seem appropriate

The model given in Fig 4 seems completely unnecessary since these are straightforward arguments.

The antitumor activity measurements are quite preliminary and may not warrant the sweeping generalizations

In summary, this is an interesting, if flawed contribution which could contribute to our understanding of how to better enhance cancer immunotherapy.

Author's Response: We are deeply appreciative of the reviewer's comments and suggestions. Below please find our response to these comments in a point-by-point manner.

Point #1: The quantitation of gels in Figures 1-4 would greatly help the reader understand the quantitative changes. Really hard to decipher what his there now.

Response to Point #1: We thank the reviewer for pointing this out. We have added quantification (fold change ratio of band intensity) in the revised manuscript. Revised Figures 4a and 4b are shown below:

Point #2: The data suggesting GSK-3 as the sole activator is lacking. This can only really be addressed by kinase array technology

Response to Point #2: We thank the reviewer for the comments. We agree with the reviewer and do not claim GSK3 β is the sole kinase that regulates PD-L1 protein. As a matter of fact, many proteins are known to be regulated by different kinase, e.g. FOXO3A^{4,5} or TSC⁶ complex which can be phosphorylated by different kinases at different amino acid residues to regulate their protein stability. Thus, other kinases may also phosphorylate PD-L1 protein and affect its stability. To avoid overstating the role of GSK3 β , we have revised the statements in page 12 to the following: "...GSK3 β as a kinase to phosphorylate PD-L1...It is known that the same protein can be regulated by multiple kinases^{4,5}. It is not yet clear whether other kinases may also phosphorylate PD-L1 to regulate its protein stability. If so, similar combination treatment strategy may be explored to improve immune check point therapy that has been shown impressive outcomes in multiple clinical trials."

Point #3: *The references seem appropriate*

Response to Point #3:

We thank the reviewer for indicating that our references seem appropriate. In the revised version, we have updated more reference to better introduce the background.

Point #4: *The model given in Fig 4 seems completely unnecessary since these are straightforward arguments.*

Response to Point #4: We agree with the reviewer and have now moved this model to Supplementary Fig 8d in the revised manuscript.

Point #5: The antitumor activity measurements are quite preliminary and may not warrant the sweeping generalizations

Response to Point #5: Thank you for the comments. To extend our findings to different cancer types, we performed same combinatory treatment in two other syngeneic animal models (EMT6 and CT26) and found that gefitinib enhanced an efficacy of anti-PD-1 antibody treatment as well. As shown in the Supplementary Fig 9, EMT6 (mouse breast cancer cells) and CT26 (mouse colon cancer cells) were injected to BALB/c mice. Tumor growth was significantly reduced when mice were treated with gefitinib and anti-PD-1 antibody. This result suggests blocking EGFR signaling enhances the efficacy of anti-PD-1 therapy in multiple types of cancer models. We have added these data and description in the revised manuscript as shown below.

In revised manuscript, page 11. We added the following: “In addition, gefitinib also enhanced an efficacy of anti-PD-1 antibody treatment in other syngeneic animal models such as EMT6 (mouse breast cancer cells) and CT26 (mouse colon cancer cells)(Supplementary Fig 9 e-g).”

Supplementary Fig 9

Reviewer #3 (Breast cancer and EGFR)(Remarks to the Author):

In this paper, Li et al. examine the molecular and biochemical mechanisms that influence the stability of PD-L1 and how that impacts the check point function of this protein. The authors have convincingly shown that PD-L1 exists in a glycosylated and non-glycosylated form, they identified the sites on the molecule that are N-glycosylated, and they showed that glycosylation has a profound effect on protein stability, with the non-glycosylated version being subjected to phosphorylation and subsequent ubiquitylation and degradation. Based on these observations, the authors propose that growth factor signaling in cancer cells represses the activity of GSK3B, the kinase that phosphorylates PD-L1 resulting in glycosylation, stabilization and immune suppression. While this is an elegant model, their data do not agree with the prediction from the model which predicts that blocking growth factor signaling can reduce the immunosuppressive function of PD-L1.

The majority of the work presented in this paper was exceptionally well done and the data qualities are outstanding in most of the figures. In some cases, appropriate statistical methods have been used, but this is not always the case. This is a very interesting and important paper, however, the authors have over-interpreted some of their results. Specific comments are below:

1. I don't understand figure 1d. I understand what the authors are trying to show, but this figure is unclear to me. By contrast, supplementary figure 1e is quite clear so I don't understand why this isn't the figure presented in the main part of the paper.
2. On page 6, the authors state "the endogenous PD-L1 turnover rate was much faster in TM treated cells than in DMSO treated cells." This needs to be more clearly stated as the figure shows that the turnover rate of only the non-glycosylated form had the faster turnover rate.
3. Next the authors state that MG132 increased non-glycosylated PD-L1 and its ubiquitination, however, this is not exactly what figure 2C shows, as there is no MG132 alone group, only cells treated with MG132 plus TM. So, that lane either needs to be added, or this sentence needs to be corrected.
4. Figure 2e shows the influence of removing glycosylation sites on PD-L1 and the effect of that on stability in the presence of CHX. It's not clear, however, why in the top three panels, two bands are shown while in the bottom four panels, only a single band is visible, and there are no markers to indicate which band we're looking at.
5. Figure 3B is entirely unclear. It is also unclear why the authors chose to show the results of a Duolink assay (I don't know what that is) instead of the more straightforward co-iPs and GST-pulldown assays that are shown in the supplement, to make this point. Once again, it seems that the supplementary data is much clearer than the data shown in the paper. Seeing dots on an IF is not very compelling. I suggest switching these out.
6. The authors point to figure 3c when they say that deletion of a region disrupts binding to GSK3B, but 3c is just a cartoon and the real data are again shown in the supplement. I recommend switching these too.
7. I am having trouble seeing the data in figure S6a on the role of the mutation in the betaTrCP destruction box and its effect on PD-L1 stability
8. The part of the paper that centers around the influence of signaling molecules and receptors on the stability of PD-L1 is the weakest part of the paper. First, it never inspires confidence when investigators keep switching cell lines in order to make specific points. The results with BT549 cells stimulated with EGF are interesting, but the A431 data are not meaningful because EGF induces profound apoptosis in this EGFR amplified cell line. Earlier in the paper, MDA-468 cells were used, which are also EGFR amplified. Why are data with that cell line not presented in this figure? Regarding figure 4b, which is connected to figure S8a, what is PVR, B7H3 and B7H4 and why are then in these figures? Furthermore, EGF did not have a consistent effect on PD-L1 levels in the other cell lines shown in figure S8a and in figure S8d, there is clearly no correlation between steady state EGFR levels and PD-L1 levels. Also, figure 4f has a model in which AKT is depicted, but there are no AKT data in this paper.
9. Figure 4e is entirely unconvincing and there is supplementary Table 1 is not interpretable as there is not enough information provided about how these data were generated, nor how the statistical analysis was done.
10. Next the authors try to make the case that the interaction between EGFR signaling and PD-L1 stability has therapeutic indications, but here again; the connection is tenuous and not convincing. The authors claim a synergistic interaction between gefitinib treatment and anti-PD1 treatment in the 4T1 tumor model. However, there is no statistical indication for synergy in the data shown. Indeed, the data shown in figure 5g shows an effect of gefitinib alone, and anti-PD1 alone on tumor growth, and the combination treatment appears to have yielded an additive response, which is not consistent with an interaction between the two drugs. In this regard, the data in figure 5i is more convincing and suggestive of an enhanced immune response following the combined treatment.
11. Because the association of PD-L1 stability with EGFR signaling is not convincing, the title of the paper needs to be changed to more accurately reflect the key findings of this paper.

In summary, this is an excellent paper on an important topic. The biochemistry is outstanding, but more work needs to be done to extend their model to tumor biology and immune therapy.

Author's Response: We are deeply appreciative of the reviewer's comments and suggestions. Below please find our response to these comments in a point-by-point manner.

Point #1: *I don't understand figure 1d. I understand what the authors are trying to show, but this figure is unclear to me. By contrast, supplementary figure 1e is quite clear so I don't understand why this isn't the figure presented in the main part of the paper.*

Response to Point #1: We thank the reviewer the comment. We agree that supplementary Figure 1e better demonstrates PD-L1 glycosylation. Following the reviewer's suggestion, we have switched original Figure 1d and *supplementary figure 1e*. The revised Figure 1 is shown below.

To show that PD-L1 is indeed glycosylated, we performed glycostaining experiment. Purified PD-L1 showed positive staining (Fig. S1e, lane 3, left panel) that was similar to the positive control (Fig. S1e, lane 1, left panel). In the presence of PNGase F,

the glycoprotein staining of PD-L1 protein was absent (Fig S1e, lane 4, left panel). The Coomassie blue panel shown on the right indicates protein loading control of the glycostaining. We also provided more information in the Material and Method section to describe the glycostaining.

Point #2: On page 6, the authors state "the endogenous PD-L1 turnover rate was much faster in TM treated cells than in DMSO treated cells." This needs to be more clearly stated as the figure shows that the turnover rate of only the non-glycosylated form had the faster turnover rate.

Response to Point #2: Thank you for this important comment. The description has been revised to the following: "the turnover rate for non-glycosylated PD-L1 was faster than glycosylated PD-L1".

In revised manuscript, Page 6, we added "the faster turnover rate of non-glycosylated PD-L1 compared with glycosylated PD-L1 was also found at endogenous PD-L1 proteins."

Point #3: Next the authors state that MG132 increased non-glycosylated PD-L1 and its ubiquitination, however, this is not exactly what figure 2C shows, as there is no MG132 alone group, only cells treated with MG132 plus TM. So, that lane either needs to be added, or this sentence needs to be corrected.

Response to Point #3: We agree with reviewer's concern and have revised our description accurately. We have rephrased our manuscript to "Non-glycosylated PD-L1 exhibited more ubiquitination in the presence of MG132 (Fig. 2c),..." in page 6.

As per reviewer's request, we have now included an ubiquitination assay showing that MG132 treatment alone did not induce PD-L1 ubiquitination (Supplementary Fig. 6f).

Supplementary Fig. 5a

Point #4: Figure 2e shows the influence of removing glycosylation sites on PD-L1 and the effect of that on stability in the presence of CHX. It's not clear, however, why in the top three panels, two bands are shown while in the bottom four panels, only a single band is visible, and there are no markers to indicate which band we're looking at.

Response to Point #4: We apologize for not describing it clearly. Two bands (or multiple bands) in the top three panels (Figure 2e) came from the heterogeneity of PD-L1 glycosylation. Two bands in the top three panels and other single band are all PD-L1 protein. The expression pattern of PD-L1 WT, N35Q and N192/200Q are consistent among Figures 1h, 2e and 2g (input section). Following reviewer's suggestion, we have added the markers to indicate the correspond molecular weight in the Figure 2e in the revised manuscript. The revised Figure 2e is shown below:

Glycosylation often results in heterogeneous expression pattern on the SDS-PAGE. The more the sites get glycosylated, the more complicated form is observed. Based on the mass spectrometry results shown in Figure 1f and supplementary Figure 4, the upper band is composed of complex type glycans (N35 and N200) and lower band is a result of high-mannose type glycosylation (N192 and N219) which is the major type of glycosylation. It is possible that N219 is the site where most high-mannose type glycosylation occurs or is critical for priming for N192 glycosylation. When N219 is mutated to Q (Gln), most of high-mannose glycosylation is lost, resulting in the disappearance of the lower band.

Fig. 1h

Fig. 2e

Fig. 2g

Point #5: Figure 3B is entirely unclear. It is also unclear why the authors chose to show the results of a Duolink assay (I don't know what that is) instead of the more straightforward co-iPs and GST-pulldown assays that are shown in the supplement, to make this point. Once again, it seems that the supplementary data is much clearer than the data shown in the paper. Seeing dots on an IF is not very compelling. I suggest switching these out.

Response to Point #5:

We agree with the reviewer that co-IP and GST-pulldown assays are clearer than Duolink data to show the protein-protein interaction. In the revised manuscript, we have moved the Duolink results to Supplementary Figure 5e. We also added detailed information in the Material and Methods section to describe how the Duolink assay was performed. The Duolink data were used to further demonstrate the endogenous PD-L1 and GSK3 β protein-protein interaction *in vivo*. Revised Figure 3 is shown below.

Point #6: The authors point to figure 3c when they say that deletion of a region disrupts binding to GSK3 β , but 3c is just a cartoon and the real data are again shown in the supplement. I recommend switching these too.

Response to Point #6: We agree with reviewer's suggestion and have switched these two figures in our revised manuscript accordingly. The IP/western blot results are now shown in Figure 3c. The entire Figure 3 has been revised as shown in the **Point #5**.

Point #7: I am having trouble seeing the data in figure S6a on the role of the mutation in the betaTrCP destruction box and its effect on PD-L1 stability

Response to Point #7:

We apologize that we did not clearly describe our data. The goal was to show that PD-L1 S176, which is located within the β -TrCP destruction box, affects β -TrCP binding in Figure S6a. First, we used PD-L1 4NQ as a model as it is degraded by

GSK3 β . We use different forms of GSK3 β and found that the higher the GSK3 β activity, the larger the difference between 4NQ and 4NQ/S176A expression. To better illustrate these results, we added a bar graph showing protein intensity of PD-L1. In the revised manuscript, we added a description in the Results section on the quantitation. Together with other *in vivo* and *in vitro* data, our findings suggest that GSK3 β mediates PD-L1 degradation via β -TrCP.

The following is shown on page 8 of the revised manuscript:

“GSK3 β -mediated PD-L1 4NQ degradation required the D/LSGxxS degron on the PD-L1. We found that GSK3 β wild type (WT) and constitutive activate GSK3 β (CA), but not kinase dead GSK3 β (KD), induced a kinase-dependent degradation”.

Supplementary Figure 6a

Point #8: *The part of the paper that centers around the influence of signaling molecules and receptors on the stability of PD-L1 is the weakest part of the paper. First, it never inspires confidence when investigators keep switching cell lines in order to make specific points. The results with BT549 cells stimulated with EGF are interesting, but the A431 data are not meaningful because EGF induces profound apoptosis in this EGFR amplified cell line. Earlier in the paper, MDA-468 cells were used, which are also EGFR amplified. Why are data with that cell line not presented in this figure? Regarding figure 4b, which is connected to figure S8a, what is PVR, B7H3 and B7H4 and why are then in these figures? Furthermore, EGF did not have a consistent effect on PD-L1 levels in the other cell lines shown in figure S8a and in figure S8d, there is clearly no correlation between steady state EGFR levels and PD-L1 levels. Also, figure 4f has a model in which AKT is depicted, but there are no AKT data in this paper.*

Response to Point #8:

We thank the reviewer for the comments. While some cell line was chosen for specific reasons, the scope of this study generally focused on the breast cancer cell lines, particularly triple-negative breast cancer cells. In the Figure 1b, we surveyed several types of cancer to show that PD-L1 glycosylation appears in four types of cancer cells, including breast, lung, and colon cancers, and melanoma. As for EGF signaling study, following the reviewer's suggestion, we have revised Figure 4a to present two breast cancer cell lines (BT549 and MB-468 cells) in response to EGF stimulation. In the revised manuscript, the study of PD-L1 from Figure 1 to 5 was all performed using the breast cancer cells. Please see revised Figure 4 below.

The inconsistencies between Figure S8a and old supplementary figure 8d was due to the experimental condition. In the supplementary figure 8a, cells were incubated in serum free medium overnight before ligand treatment whereas the old Figure S8d shows the basal level of PD-L1 expression in normal culture medium with serum. We have now included that in the figure and legend in the revised manuscript to better distinguish the two.

Regarding Figure 4b, PVR, B7H3, and B7H4 are known ligands of immune receptors and expressed in tumor cell as PD-L1. In Figure 4b and S8a, EGF treatment upregulated only PD-L1 but not other ligands such as PVR, B7H3, and B7H4. These data suggest that EGF/EGFR/GSK3 β /PD-L1 regulation axis is a specific for PD-L1 but not ligands for other immune receptors. We have updated the description to clearly cite Figure 4b and Supplementary Figure 8a on page 10 of the revised manuscript as follows:

“Among those examined, only EGF specifically induced PD-L1 expression. Other ligands for other immune receptors such as PVR, B7H3, and B7H4 did not respond to EGF in two tested cell lines (Figs 4a, top, 4b, and Supplementary Fig. 8a).”

Based on the reviewer’s suggestion, we have removed AKT from our model. We are also aware that supplementary figure 8b is not necessary because EGF treatment data is mostly overlapped with supplementary figure 8a. Since IFN γ is not main focus of current study, we have removed the old Supplementary figs. 8b and 8d from the revised manuscript to avoid confusion.

Figure 4

Point #9: Figure 4e is entirely unconvincing and there is supplementary Table 1 is not interpretable as there is not enough information provided about how these data were generated, nor how the statistical analysis was done.

Response to Point #9:

We apologize for not describing the IHC results clearly. We have added detailed description in the figure legend and Methods and also modified supplementary Table 1 as shown below. In addition, we have moved the original figure 4e to Supplementary Fig 8c in the revised manuscript. Please see figure 4 in our response to **Point #8**.

Revised supplementary Figure 8c legend:

“Representative images from IHC staining of p-EGFR (phosphorylation of Tyr 1068, a marker of EGFR activation), p-GSK3b (phosphorylation of Ser 9, a marker of inactivation of GSK3b), PD-L1, and granzyme B (a marker of cytotoxic T cell activation) in primary breast tumor tissues.”

Methods

“Immunohistochemical staining of human breast tumor tissue samples

Human breast tumor tissue specimens were obtained following the guidelines approved by the Institutional Review Board at MD Anderson Cancer Center, and written informed consent was obtained from patients in all cases at time of enrollment. Immunohistochemical (IHC) staining was performed as described previously^{7,6}. Briefly, tissue specimens were incubated with antibodies against PD-L1 (Cell Signaling Technology, #13684, 1:30 dilution), p-EGFR (Y1068) (#3777, Cell Signaling Technology, 1: 200 dilution), p-GSK3 β (S9) (#9323, Cell Signaling Technology, 1:200 dilution) or Granzyme B (ab4059, Abcam, 1:100 dilution) and a biotin-conjugated secondary antibody and then incubated with an avidin-biotin-peroxidase complex. Visualization was performed using amino-ethylcarbazole chromogen. For statistical analysis, Fisher’s exact test and Spearman rank correlation coefficient were used and a p-value less than 0.05 is considered statistically significant. The intensity of staining was ranked into four groups: high (+++), medium (++), low (+), and negative (-) according to histological scores.”

Supplementary Table 1.

Supplementary Table 1. Correlations between expression levels of PD-L1, p-EGFR (Tyr 1068), p-GSK3 β (Ser 9), and granzyme B expression in surgical specimens of breast cancer.

		No. expression of PD-L1 (%)			P value
		- / +	++ /+++	Total	
p-EGFR	- / +	26 (45.6%)	28 (25.2%)	54 (32.1%)	P = 0.007
	++ /+++	31 (54.4%)	83 (74.8%)	114 (67.9%)	
	Total	57 (100%)	111 (100%)	168 (100%)	
p-GSK3 β	- / +	27 (42.9%)	4 (5.2%)	31 (22.1%)	P = 0.0001
	++ /+++	36 (57.1%)	73 (94.8%)	109 (77.9%)	
	Total	63 (100%)	77 (100%)	140 (100%)	
Granzyme B	- / +	50 (68.5%)	92 (81.4%)	142 (76.3%)	P = 0.043*
	++ /+++	23 (31.5%)	21 (18.6%)	44 (23.7%)	
	Total	73 (100%)	113 (100%)	186 (100%)	

P, Spearman correlation; *, An inverse correlation between PD-L1 and granzyme B was found. -/+, negative or low expression; ++/+++, medium or high expression.

Point #10: Next the authors try to make the case that the interaction between EGFR signaling and PD-L1 stability has therapeutic indications, but here again; the connection is tenuous and not convincing. The authors claim a synergistic interaction between gefitinib treatment and anti-PD1 treatment in the 4T1 tumor model. However, there is no statistical indication for synergy in the data shown. Indeed, the data shown in figure 5g shows an effect of gefitinib alone, and anti-PD1 alone on tumor growth, and the combination treatment appears to have yielded an additive response, which is not consistent with an interaction between the two drugs. In this regard, the data in figure 5i is more convincing and suggestive of an enhanced immune response following the combined treatment.

Response to Point #10:

We agree with reviewers' comment and following the suggestion, we have changed "gefitinib treatment enhances the efficacy of anti-PD-1 antibody treatment" instead of "the synergistic interaction" in our revised manuscript at several places as shown below. In the revised manuscript, we also generalized our animal study using additional mouse breast cancer cells (EMT6) and mouse colon cancer cells (CT26). The data are consistent and suggest that gefitinib treatment enhances anti-PD-1 therapy (Supplementary Fig 9e-g). We also strengthen our rationale for the combinatory therapy. Because PD-L1 also binds to CD80 for T cell suppression^{1,2}. To block PD-L1/PD1 as well as other possible ligand/receptor binding such as PD-L1/CD80 axis effectively, we combined gefitinib and anti-PD-1 antibody treatments.

On page 2:

"Inhibition of EGF signaling by gefitinib destabilizes PD-L1, enhances antitumor T cell immunity and the therapeutic efficacy of anti-PD-1 in syngeneic mice models."

On page 10:

"The proposed model prompted us to test whether inhibition of EGF-mediated PD-L1 stabilization enhances blockade of PD-L1/PD1 therapy.

On page 11:

"Consistent with our observations *in vitro*, gefitinib enhanced the efficacy of the PD-1 antibody efficacy in a 4T1-luciferase (4T1-Luc) syngeneic BALB/c model."

"Together, inhibition of PD-L1 stabilization by gefitinib enhances an efficacy of anti-PD-1 in syngeneic mice models."

On page 12:

"Importantly, inhibition of EGF-mediated PD-L1 stabilization enhances the therapeutic efficacy of anti-PD-1 to promote tumor-infiltrating cytotoxic T cell immune response."

Supplementary Fig. 9

Point #11: Because the association of PD-L1 stability with EGFR signaling is not convincing, the title of the paper needs to be changed to more accurately reflect the key findings of this paper.

Response to Point #11:

Following the reviewer's suggestion, we have changed our title to "**Glycosylation and stabilization of PD-L1 suppresses T cell activity**" to emphasize the key findings on the molecular mechanism regulating PD-L1 stability.

References:

1. Butte, M.J., Keir, M.E., Phamduy, T.B., Sharpe, A.H. & Freeman, G.J. Programmed death-1 ligand 1 interacts specifically with the B7-1 costimulatory molecule to inhibit T cell responses. *Immunity* 27, 111-122 (2007).
2. Yang, J. *et al.* The novel costimulatory programmed death ligand 1/B7.1 pathway is functional in inhibiting alloimmune responses in vivo. *Journal of immunology* 187, 1113-1119 (2011).
3. Kleffel, S. *et al.* Melanoma Cell-Intrinsic PD-1 Receptor Functions Promote Tumor Growth. *Cell* 162, 1242-1256 (2015).
4. Yang, J.Y. *et al.* ERK promotes tumorigenesis by inhibiting FOXO3a via MDM2-mediated degradation. *Nature cell biology* 10, 138-148 (2008).
5. Yamaguchi, H., Hsu, J.L. & Hung, M.C. Regulation of ubiquitination-mediated protein degradation by survival kinases in cancer. *Frontiers in oncology* 2, 15 (2012).
6. Lee, D.F. *et al.* IKK beta suppression of TSC1 links inflammation and tumor angiogenesis via the mTOR pathway. *Cell* 130, 440-455 (2007).
7. Li, C.W. *et al.* AKT1 Inhibits Epithelial-to-Mesenchymal Transition in Breast Cancer through Phosphorylation-Dependent Twist1 Degradation. *Cancer Res* 76, 1451-1462 (2016).

Reviewers' Comments:

Reviewer #1 (Remarks to the Author)

No further comments on the manuscript

Reviewer #2 (Remarks to the Author)

Authors have addressed my major concerns.

Reviewer #3 (Remarks to the Author)

This version of the paper is significantly stronger than the original version and I am satisfied with the authors response to my review. At this time, I only have to minor comments. First, while the authors have shown evidence that in some model systems, EGF or other EGF-like ligands influence expression of PD-L1 and in that way influence immune response, their data in supplementary table 1 shows that while there is a statistically significant correlation between presence of active EGFR and expression of PD-L1, the majority of samples with low PD-L1 had high p-EGFR. So, I think the connection to EGFR signaling needs to be downplayed a bit, as this mechanism seems to be operative in a subset of EGFR-positive patients. Second, while I appreciate the authors removing AKT from their model, they still discuss its possible role at the end of the paper, which concerns me since there are no data on AKT anywhere in the paper. It seems to me that it would have been just as easy to look at the influence of an AKT inhibitor on PD-L1 expression as it was to look at gefitinibs effects, yet that experiment wasn't done or the results were not consistent with the model. Given that, I'd really like to see some statement that indicates that none of their data implicate AKT activation in the mechanism they've demonstrated in this paper.

Point-by-Point Response Letter

Reviewers' Comments

Reviewer #2 (Remarks to the Author): *Authors have addressed my major concerns.*

Authors' Response: We sincerely thank the reviewer for his/her time in reviewing our manuscript.

Reviewer #3 (Remarks to the Author): *This version of the paper is significantly stronger than the original version and I am satisfied with the authors response to my review. At this time, I only have to minor comments. First, while the authors have shown evidence that in some model systems, EGF or other EGF-like ligands influence expression of PD-L1 and in that way influence immune response, their data in supplementary table 1 shows that while there is a statistically significant correlation between presence of active EGFR and expression of PD-L1, the majority of samples with low PD-L1 had high p-EGFR. So, I think the connection to EGFR signaling needs to be downplayed a bit, as this mechanism seems to be operative in a subset of EGFR-positive patients. Second, while I appreciate the authors removing AKT from their model, they still discuss its possible role at the end of the paper, which concerns me since there are no data on AKT anywhere in the paper. It seems to me that it would have been just as easy to look at the influence of an AKT inhibitor on PD-L1 expression as it was to look at gefitinib effects, yet that experiment wasn't done or the results were not consistent with the model. Given that, I'd really like to see some statement that indicates that none of their data implicate AKT activation in the mechanism they've demonstrated in this paper.*

Authors' Response: We thank the reviewer for his/her time in reviewing our manuscript and appreciate the additional comments. We agree with reviewer's suggestion. In the revised manuscript, we have toned down the positive correlation between p-EGFR and PD-L1 to suggest it may be specific to a certain subset of EGFR-positive patients as shown on page 11 of the revised manuscript:

“Of note, the majority of samples with low PD-L1 had high p-EGFR expression. Therefore, EGFR-mediated PD-L1 stabilization may appear in a subset of EGFR-positive patients.”

With regard to the role of AKT, because we do not have data to show that AKT is directly involved in EGF-mediated PD-L1 stabilization, we have now rephrased the Discussion on page 13 as shown below:

“Thus far, we do not know whether AKT directly regulates PD-L1 expression. More in-depth analysis would be required to justify the role of AKT in EGFR-mediated PD-L1 protein stabilization.”